# Decoding the molecular, cellular, and functional heterogeneity of zebrafish intracardiac nervous system

Andrea Pedroni[1,4], Elanur Yilmaz [2,3,4], Lisa Del Vecchio[1], Prabesh Bhattarai [2,3], Inés Talaya Vidal[1], Yu-Wen E. Dai[1], Konstantinos Koutsogiannis[1], Caghan Kizil [2,3] ✉ & Konstantinos Ampatzis [1] ✉

The proper functioning of the heart relies on the intricate interplay between the central nervous system and the local neuronal networks within the heart itself. While the central innervation of the heart has been extensively studied, the organization and functionality of the intracardiac nervous system (IcNS) remain largely unexplored. Here, we present a comprehensive taxonomy of the IcNS, utilizing single-cell RNA sequencing, anatomical studies, and electrophysiological techniques. Our findings reveal a diverse array of neuronal types within the IcNS, exceeding previous expectations. We identify a subset of neurons exhibiting characteristics akin to pacemaker/rhythmogenic neurons similar to those found in Central Pattern Generator networks of the central nervous system. Our results underscore the heterogeneity within the IcNS and its key role in regulating the heart's rhythmic functionality. The classification and characterization of the IcNS presented here serve as a valuable resource for further exploration into the mechanisms underlying heart functionality and the pathophysiology of associated cardiac disorders.

Maintaining cardiovascular homeostasis is crucial for animal survival. Therefore, cardiac output must be quickly adjusted to meet the tissue's blood perfusion demands. Complex interactions between the parasympathetic and sympathetic components of the autonomic nervous system (ANS) maintain cardiovascular homeostasis by exerting opposing effects on all cardiac functional indices (chronotropy, dromotropy, inotropy, and lusitropy)[1]. The innervation of the heart is complex and compromised by the central, extracardiac, and intrinsic cardiac components of the ANS[1–4].

The intracardiac nervous system (IcNS), which is embedded within the superficial layers of the heart wall, has long been considered a simple parasympathetic postganglionic structure that relays central efferent information[2,3,5–7]. However, over the past few decades, studies have started uncovering the intricate structural and functional organization of the IcNS. Consequently, it has been proposed to be the ultimate final hub for signal integration in the cardiac autonomic nervous system, which has led to the notion that it acts as the heart's "little brain"[8–10]. In support of this model, the IcNS involves an interconnected cluster of different parasympathetic and sympathetic neurons, sensory neurons, local regulatory interneurons, and motor neurons that play a pivotal role in the local neural control of cardiac performance in the absence of input from the central nervous system[11]. While the IcNS contributes to several cardiac parameters like heart contractility, rate, and conduction velocity[7,12,13], its rythmogenic ability remains elusive.

Here, we combined a methodological approach integrating single-cell RNA sequencing, neurochemistry, and electrophysiology to characterize adult zebrafish intracardiac neurons, an animal model

[1]Department of Neuroscience, Karolinska Institutet, 171 77 Stockholm, Sweden. [2]Taub Institute for Research on Alzheimer's Disease and the Aging Brain, Vagelos College of Physicians and Surgeons, Columbia University Irving Medical Center, Columbia University, New York, NY 10032, USA. [3]Department of Neurology, Columbia University Irving Medical Center, Columbia University, New York, NY 10032, USA. [4]These authors contributed equally: Andrea Pedroni, Elanur Yilmaz. ✉e-mail: ck2893@cumc.columbia.edu; Konstantinos.Ampatzis@ki.se

that exhibits strong similarities to human heart rate and overall cardiac functionality[14–16]. Our experimental strategy yielded an enhanced structural and functional map of neuronal diversity within the IcNS. The comprehensive functional studies using our ex-vivo preparation of the adult zebrafish heart revealed a small group of intracardiac neurons with pacemaker/rhythmogenic properties. Collectively, our findings support the crucial role of the IcNS in heart functionality and provide a foundation for a further understanding of how different neuronal types within the IcNS are involved in various physiological and pathological conditions.

## Results

### Neuroanatomy of zebrafish intracardiac nervous system

The zebrafish has a prototypic vertebrate heart that consists of four chambers (sinus venosus, atrium, ventricle, and bulbus arteriosus) connected sequentially through valves[15,17] (Fig. 1a). We first sought to map the innervation pattern of the adult zebrafish heart and to define the anatomical distribution of the intracardiac neurons by combined immunolabeling against the pan-neuronal markers HuC/D, NeuN, and the general marker of neuronal processes Zn-12, (Fig. 1b and Supplementary Fig. 1). We observed that both HuC/D and NeuN labeled neuronal somata predominantly near the valves connecting the different heart compartments (Fig. 1b). However, the NeuN could mark 87.9% of the HuC/D+ neurons, and we did not observe any NeuN+HuC/D- neurons (Supplementary Fig. 1). Thus, we focused on the expression of HuC/D as a marker of most of the intracardiac neuron somata. HuC/D+ neurons were present in all heart areas, displaying varying sizes (Fig. 1b–d). While most intracardiac neurons (91.7%) were positioned near the valves and were in close contact with the Zn-12+ processes, the largest group of neurons (74.1%) was detected at the sinoatrial valve area forming the sinoatrial plexus (SAP) as described earlier in teleost

fish, including zebrafish[17,18] (Fig. 1b, c). The detected neurons of the zebrafish SAP area (extended 80μm from the epicenter of the valve) consisted of a population of 81 ± 3.6 HuC/D+ neurons with a wide range of soma sizes without an apparent topography (Fig. 1e, f), indicating the possibility that it contains neurons with discrete molecular, biochemical, and functional properties.

### Molecular landscape of the Intracardiac Nervous System

To perform an unbiased characterization of the cell types of the zebrafish heart and to identify the molecular signatures of the intracardiac neurons, we performed single-cell RNA sequencing (scRNA-seq; two replicated experiments) from cells derived from the adult zebrafish heart using the *Tg(elavl3:EGFP)* and Tg(nbt:DsRed) animal lines. The datasets from both scRNA-seq were integrated for an overall unified analysis. After quality control, we obtained and processed 9508 heart cells (Fig. 2a–c, Supplementary Fig. 2). Following data integration and unsupervised dimensionality reduction approaches revealed 22 transcriptionally distinct cell clusters forming 8 separate cell populations (Fig. 2a,b). While cardiomyocytes (CM) made up the greatest proportion of the heart cells, we were also able to detect a small fraction of neuronal cells (NE; cluster #17) based on the canonical neuronal markers *elav3* and *sv2a* (Fig. 2b). The population of neuronal cells expressed several selected neuronal marker genes, along with genes encoding different neurotransmitter receptors, such as *chrm2a, chrna3, and chrna5* (muscarinic and nicotinic cholinergic receptors), *gabbr1a* and *gabrg2* (GABA receptors), *glra1* and *glrbb* (glycine receptors), *gria2a, grid2, grik4* and *grm1a* (ionotropic and metabotropic glutamate receptors) *adra2a* (adrenergic receptors) and *htr1fa* (serotonergic receptors) (Fig. 2d, e), suggesting a complex and diverse neurotransmission within the intracardiac nervous system. In addition, neurotransmitter receptor gene expression was also identified in

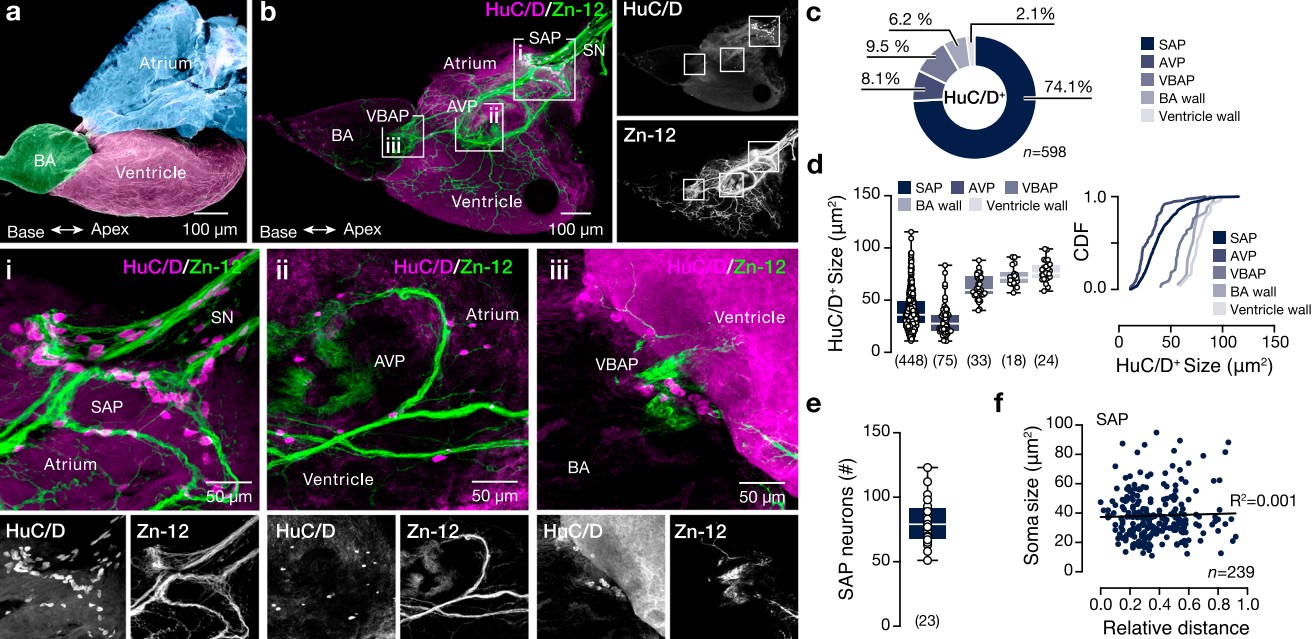

**Fig. 1 | Neuroanatomy of the adult zebrafish intracardiac nervous system.**
**a** Color-coded presentation of the major regions of the adult zebrafish prototypic heart. **b** A representative whole-mount photomicrograph shows the general innervation pattern of the adult zebrafish heart with Zn-12 immunostaining (neuronal processes) and HuC/D (neuronal somata). The regions around the valves connecting the heart chambers hold the majority of the observed HuC/D+ neuronal somata (inserts i-iii). **c** Quantification of the proportion of HuC/D+ neurons found in different heart regions. **d** Regionalized quantification of the HuC/D+ cell soma size in the adult zebrafish heart. **e** Quantification of the total number of the HuC/D

labeled neurons in the SAP area. **f** Analysis of the neuronal soma size in relation to relative distance from the sinoatrial valve. AVP atrioventricular plexus, BA bulbus arteriosus, CDF cumulative distribution frequency, HuC/D elav3&4, SAP sinoatrial plexus, SN sinus venosus, VBAP ventriculo-bulbus arteriosus plexus, Zn-12 neuronal cell surface marker (HNK-1). Data are presented as box plots showing the median with 25/75 percentile (box and line) and minimum-maximum (whiskers) and as mean ± SEM. For detailed statistics, see Supplementary Table 1. Source data are provided as a Source Data file.

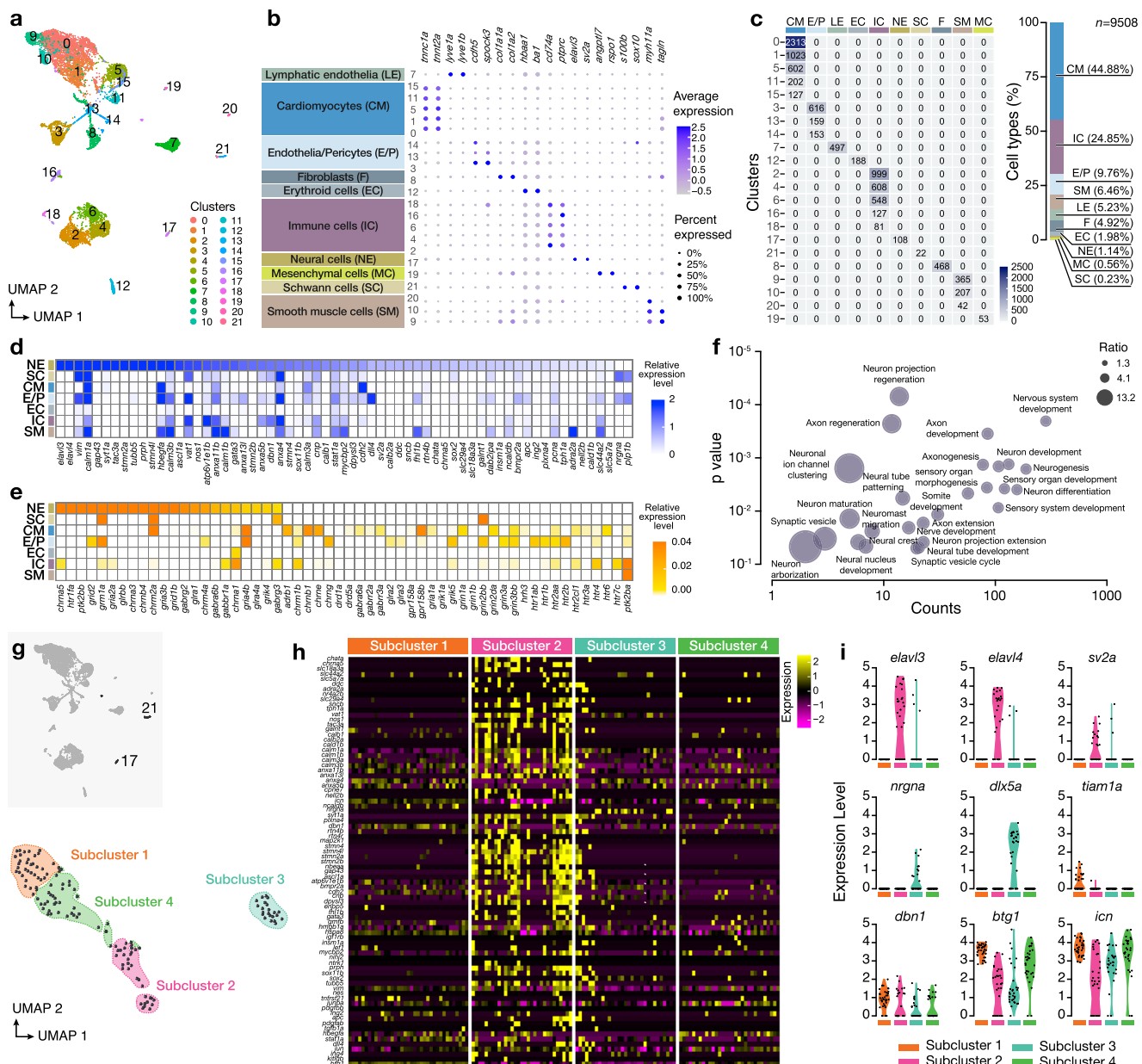

**Fig. 2 | Single-cell transcriptomics from the adult zebrafish heart. a** UMAP clustering plot identifies 22 cell distinct clusters colored differentially. **b** Dot plot of the average expression level in different cell clusters (color) and percentage of expressing cells (circle size) of selected marker genes for major cell populations (columns) across the adult zebrafish heart cells. **c** Heatmap for the number of cells sequenced per cell cluster. A bar plot showing the percent of cell types in single-cell samples. **d** Heatmap shows the expression of selected neuronal marker genes. **e** Heatmap presentation of the expression of neurotransmitter receptor genes in the neuronal cluster (NE) and non-neuronal clusters. **f** Selected gene ontology (GO) terms for neuronal cluster compared to other cell types in a multiple variable graph. The Y-axis indicates p values, and the X-axis shows the count number of genes in respective GO terms. Genes were identified through differential gene expression analyses. Bubble size indicates the odds ratio. **g** Subclustering of neurons (Cluster #17 and #21) identifies four subclusters (1 to 4). **h** Heatmap depicting the expression levels of a selected list of neuronal markers in all neuronal-related subclusters (from clusters #17 and #21). Every column indicates a cell, and every row represents a gene expression. The different neuronal subclusters are indicated and color-coded. **i**. Violin plots of the expression of selected genes in each distinct subcluster (color-coded).

various clusters of heart cells, implying significant interactions between the intracardiac and extracardiac nervous systems with cardiomyocytes, immune cells, endothelial cells, and Schwann cells (Fig. 2e).

Next, we analyzed the neuronal cluster based on biological Gene Ontology (GO) terms. Most GO terms were related to neuronal development, ion channels, synaptic formation and function, and neuronal regeneration (Fig. 2f), confirming the diverse functional states of the intracardiac neuronal population.

To further explore the heart neuronal-related population more thoroughly, we subclustered all neuronal cells together with Schwann cells (cluster #17 and #21, respectively) within the dataset and identified four transcriptionally distinct neuronal-related cell subpopulations (Fig. 2g, h and Supplementary Fig. 3). Subclusters 1 and 4 expressed a varying combination of three epithelial genes *icn, vim*, and *anxa5*[19–22] (Fig. 2h and Supplementary Fig. 4b), classifying them as neural epithelia. In addition, subclusters 1 and 4 expressed *btg1* (Anti-Proliferation Factor 1), a gene highly expressed in adult neurogenic

niches in the CNS, promoting the stem cell quiescent state[23,24], and *dbn1* (drebrin 1) that play a key role in brain development and regulation neural stem cells[25] (Fig. 2i), suggesting the presence of cells under development or with neurogenic potential. Subcluster 3 highly expressed *sox10* and *dlx5a*, two genes marking Schwann cell lineage[26,27] (Fig. 2h and Supplementary Fig. 4c), defining them as the Schwann cell cluster with specific molecular signatures (Supplementary Dataset 2). Subcluster 2 was enriched in the expression of *sv2a* (synaptic vesicle glycoprotein 2 A) and *elav3* and *elav4* markers of mature (post-mitotic) neurons (Fig. 2i), corresponding to the previously described HuC/D population (Fig. 1). Subcluster 2 also expressed *ascl1a* and *gap43*, markers of neural cells[28,29] (Supplementary Fig. 4d). Neuronal clusters had distinct gene expression signatures compared to all other cell types in the heart (Supplementary Dataset 3). Additionally, Subcluster 2 expresses different neurotransmitter receptors, indicative of heterogenous neuronal subtypes (Supplementary Fig. 4e). Together, our data suggest that although heart neuronal-related cells represent less than 1.5% of heart cells, they exhibit diverse molecular signatures, including progenitor epithelial state to neuronal and Schwann cell signatures. Thus, a comprehensive biochemical and electrophysiological classification could provide valuable context to our scRNA-seq findings.

## Neurotransmitter typology map of the Intracardiac Nervous System

Because single-cell sequencing data are inherently sparse and, thus, often have limited sensitivity to detect transcript abundance, we leveraged immunodetection approaches to contextualize our findings regarding the presence of biochemically different SAP neuronal types in the adult zebrafish heart. Based on the canonical markers *elav3* (HuC) and *elav4* (HuD), neuronal cells captured in subcluster 2 and some in subcluster 3 with scRNA-seq belong to previously presented postmitotic/mature neurons (Fig. 1b); thus, we focused our follow-up analysis on the HuC/D neuronal population located in the SAP region. Previous studies have highlighted the cholinergic nature of the vertebrate intracardiac neurons[11,17,18,30–35]. Hence, we first sought to examine the cholinergic phenotype of the SAP neurons. We observed the presence of multiple ChAT$^+$ somata and processes embracing the sinoatrial valve area (Fig. 3a, b). While the majority (80.78%) of the neurons were cholinergic (HuC/D$^+$ChAT$^+$), we also detected a population of non-cholinergic neurons (HuC/D$^+$ChAT$^-$) that had overall smaller soma sizes (Fig. 3c, i), indicating the presence of neurons of a different neurotransmitter phenotype. Therefore, we attempted to create a detailed map of the neurotransmitter typology of the adult zebrafish SAP neurons by investigating the presence of classical neurotransmitters. We observed the presence of numerous catecholaminergic (TH$^+$) neuronal processes in all heart compartments, with the vast majority of these existing in the bulbus arteriosus (BA) area. The SAP neurons were found to receive strong catecholaminergic (TH$^+$) input; however, only a few TH$^+$ neurons (4.6%) were identified (Fig. 3d, i). Moreover, the strong expression of the α2B adrenoceptors on the SAP neurons indicated the adrenergic nature of the catecholaminergic innervation (Fig. 3e). We also observed a small population of serotonergic (5-HT$^+$; 5%), glutamatergic (vGlut1$^+$ & vGlut2a$^+$; 8%), and GABAergic (GABA$^+$ & Gad1b$^+$; 6%) neurons (Fig. 3f–h, i). Soma size measurements showed that catecholaminergic, serotonergic, glutamatergic, and GABAergic neurons had similar mean soma sizes and significantly differed from the cholinergic population (Fig. 3i). The cholinergic, glutamatergic, and part of the GABAergic neurons had similar distribution patterns in proximity to the sinoatrial valve, whereas serotonergic, catecholaminergic, and another part of GABAergic neurons detected more distal (Fig. 3j). Collectively, our findings reveal a neurochemically complex and heterogeneous orchestration of the adult zebrafish SAP, complementing previous studies[17,36].

## Electrical properties of the intracardiac neurons

The study of the electrophysiological properties of the vertebrate IcNS neurons using isolated in-vitro preparations[31,37–40] could alter their passive and active electrical properties and cannot detect synaptic events. To overcome this issue, we developed an intact whole-mount ex-vivo adult zebrafish heart preparation, allowing us to thoroughly examine the firing, electrical, and synaptic properties of the intact SAP neurons integrated into the intracardiac network (Fig. 4a). We observed that repetitive firing varied significantly between the SAP neurons in response to increasing depolarizing current steps (Fig. 4b). Using the elicited firing responses to the current injection steps, we categorize the SAP neuron into four broad firing classes. Single Spike type (S) displayed strong spike frequency adaptation, always firing a single action potential; Adaptive type (A) discharged several action potentials with pronounced spike frequency adaptation; Repetitive type (R) fired during the whole depolarizing current pulse duration without spike frequency adaptation; and Bursting type (B) that fired in bursts (Fig. 4b). Moreover, the bursting SAP neurons displayed a profound SAG potential and post-inhibitory rebound firing, properties that are also present after the Na$^+$ channel blocker tetrodotoxin (TTX; Fig. 4b and Supplementary Fig. 5) application and are associated with the pacemaker neurons[41,42]. Random recordings of several SAP neurons ($n = 33$) revealed the differential representation of the four SAP neuronal types (Fig. 4c). A comprehensive analysis exposed specific cellular properties distinctive to the four adult SAP neuron type categories (Fig. 4d and Supplementary Fig. 6). We found that the resting membrane potential (RMP), the input resistance, the rheobase, the firing threshold, the AP peak and amplitude, the AP half-width duration, and the SAG amplitude were significantly different, separating the SAP neuron types, especially the Single Spike type with the Bursting type (Fig. 4d and Supplementary Fig. 6). Recording the spontaneous activity of the SAP neurons, we observed differences in firing probability between the different types. We found that the single spike type displayed no spontaneous firing, and the adaptive and repetitive type fired occasionally (Fig. 4e). However, the bursting type always fired in a regular or irregular pattern (Fig. 4e). We next tested whether the different types of SAP neurons receive different synaptic drive, by recording the excitatory post-synaptic currents (EPSCs) at voltage-clamp mode (Fig. 4f). We observed no differences in the frequency or amplitude of EPCSs (Fig. 4f).

Our analysis was largely based on supervised criteria regarding the firing pattern, and each electrophysiological property was evaluated separately from the others. Hence, to confirm the presence of functionally distinct types of SAP neurons in the adult zebrafish heart, we employed the Principal Component Analysis (PCA) to reduce the dimensionality of the cellular properties in an unsupervised manner (Fig. 4g). The structure of the PCA distribution separated the SAP neurons into three clusters. The Single Spike type and the Bursting type were separated from the rest; however, the Adaptive type and the Repetitive type were intermingled, suggesting that they share similar characteristics and potentially form subpopulations of the same group of SAP neurons (Fig. 4g). Collectively, our findings indicate that there is a diverse population of SAP neurons organized into physiologically distinct types. Importantly, we discovered the Bursting SAP neurons that display several hallmarks of pacemaker neurons, indicating that the SAP network may be capable of not only regulating but also generating the heart rhythm.

## Intracardiac neurons affect the heart's functional parameters

Motivated by our previous findings, we sought to probe basic functional associations between the intracardiac neurons and the heart functionality. We performed whole-cell patch-clamp recordings on individual atrial cardiomyocytes close to the SAP area using the ex-vivo adult zebrafish heart preparation (Fig. 5a, b). We observed that the cardiomyocyte firing frequency was correlated with the heartbeat rate

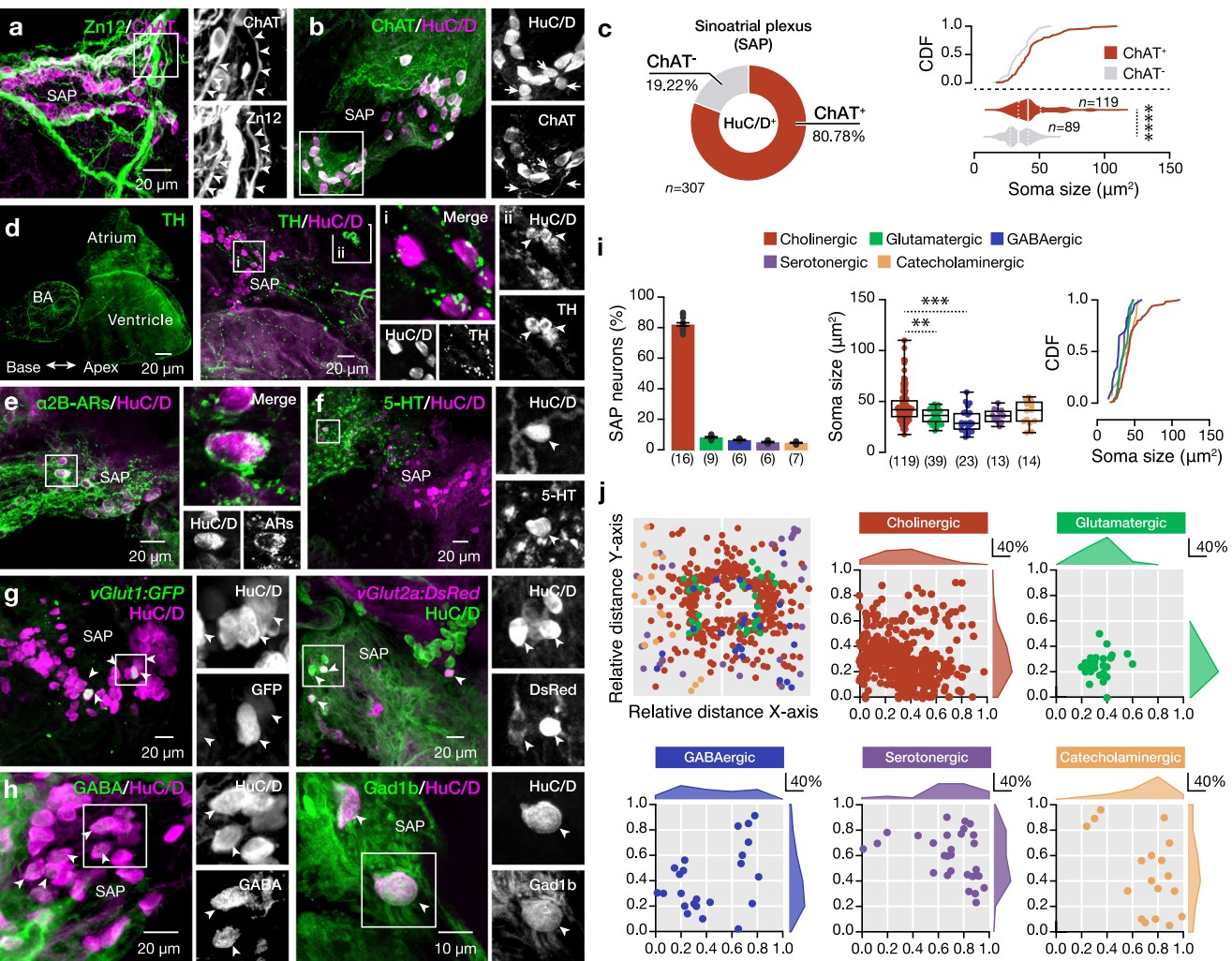

**Fig. 3 | Neurochemical variability in the intracardiac SAP neuronal population.**
**a** Representative whole-mount photomicrograph showing the abundance of cholinergic (ChAT+, magenta) neurons and processes in close contact with processes (Zn12+, green) innervating the SAP area. Arrowheads indicate the ChAT+/ Zn12+ neuronal processes. **b, c** Microphotograph showing that the majority but not all of the SAP neurons (HuC/D+) are cholinergic (ChAT+). Quantification of the proportion of the HuC/D+ that are cholinergic and analysis of the cholinergic (ChAT+) and non-cholinergic (ChAT-) SAP neuron soma sizes. Arrows indicate the non-cholinergic SAP neurons (HuC/D+/ ChAT-). **d** Whole-mount image shows the TH expression pattern on the adult zebrafish heart. Most SAP neurons (magenta) are in contact with TH+ processes (green) (in *i*). A small cluster of TH-expressing neurons in *ii*. **e** HuC/D+ neurons and processes in SAP express the adrenoceptor a2B, indicating the presence of strong adrenergic innervation. **f** Presence of serotonin (5-HT+) expressing neurons. **g** Identification of glutamatergic neurons (vGlut1+ and

vGlut2a+) in SAP neuronal population. **h** SAP contains some GABAergic (GABA+ and Gad1b+, green) neurons. **i.** Analysis of the proportion of SAP neurons expressing different neurotransmitters. Quantification and cumulative frequency of the neurochemically distinct neuron soma sizes. **j** Spatial distribution pattern of neurons expressing different neurotransmitters in SAP area. Arrowheads indicate the double-labeled neurons. 5-HT 5-hydroxytryptamine/serotonin, AR adrenergic receptor, CDF cumulative distribution frequency, ChAT choline acetyltransferase, GABA γ-Aminobutyric acid, Gad1b Glutamate decarboxylase 1, HuC/D elav3 + 4, SAP sinoatrial plexus, TH Tyrosine Hydroxylase, vGlut vesicular glutamate transporter, Zn12 neuronal cell surface marker (HNK-1). Data are presented as mean ± s.e.m., as violin plots, or as box plots showing the median with 25/75 percentile (box and line) and minimum–maximum (whiskers). ****P < 0.0001. For detailed statistics, see Supplementary Table 1. Source data are provided as a Source Data file.

obtained from video recordings of the ex-vivo heart preparation (Fig. 5c). Next, we examined whether the heart's contractile properties could be impacted by the uncoordinated synaptic release (depletion) from intracardiac neurons' readily releasable pool[43–45]. We verified that bath application of sucrose (240 mM) significantly increased the number of the recorded post-synaptic current events (EPSCs) in all examined intracardiac neurons (Fig. 5d). Accordingly, sucrose-mediated synaptic vesicle release reduced the firing frequency and regularity of all recorded cardiomyocytes (Fig. 5e, f). To test if the observed cardiomyocyte firing disruption affects the heartbeat rate, we performed similar experiments using video recordings, and we found that bath application of sucrose solution significantly reduced the heartbeat rate, yet it could not eradicate it (Fig. 5g). Finally, to confirm that the SAP neurons mediate the observed heart frequency

reduction, we developed a reduced version of the heart ex-vivo preparation by dissecting the sinus venosus/SAP area in which the heart rate was unaffected (Fig. 5h). We found that the application of sucrose solution in reduced preparation did not alter the heart frequency (Fig. 5i), implying that intracardiac neuron release is the primary source of observed disruption in frequency and regularity. Our experiments suggest that the IcNS plays a significant role in setting and modulating the cardiac rhythm by directly affecting the nodal peace-making system.

## Discussion

We have conducted a comprehensive molecular and functional classification of adult zebrafish intracardiac neurons, revealing an intricate diversity and complexity in the orchestration of the embedded

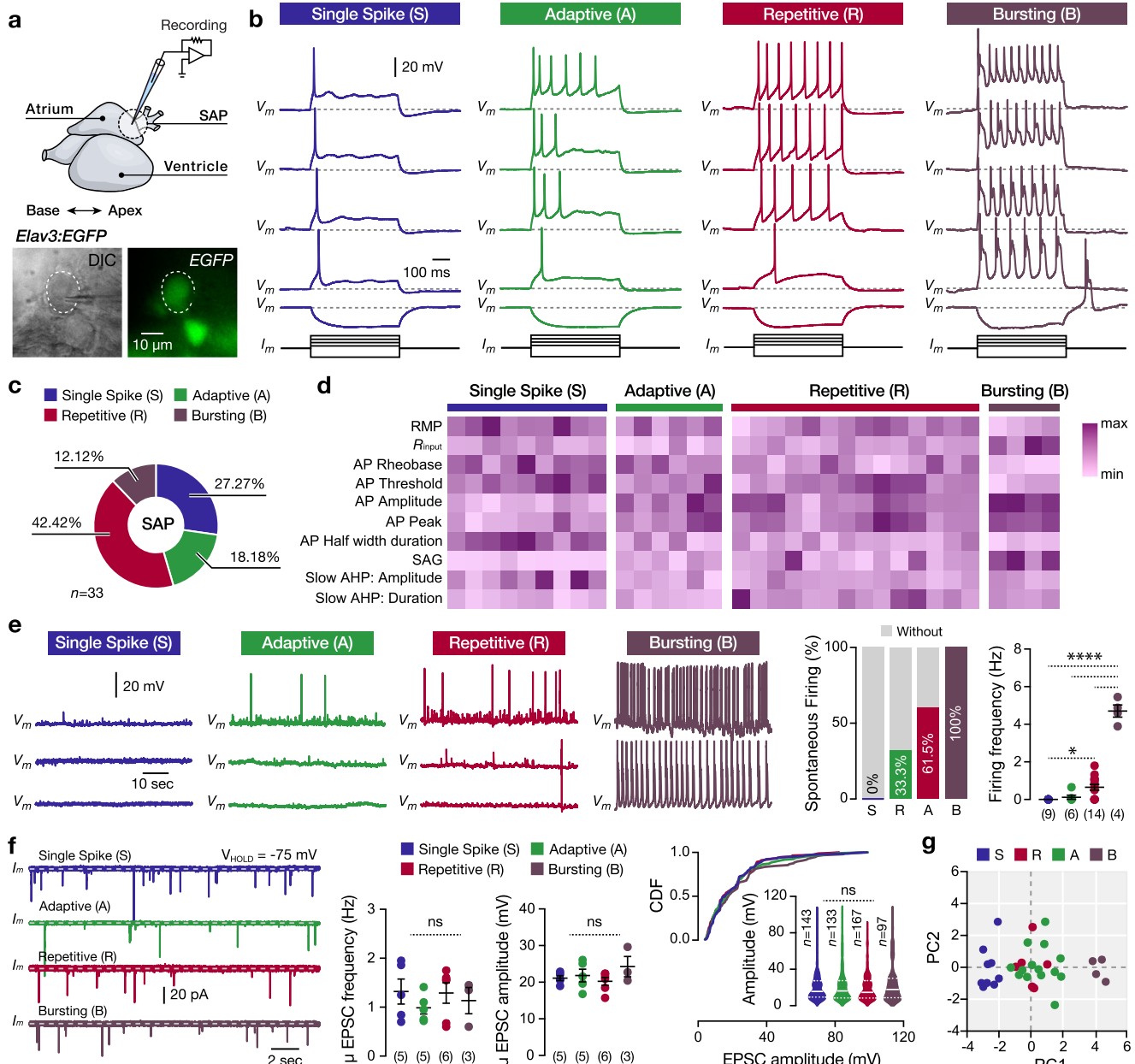

**Fig. 4 | Diverse cellular properties of the intracardiac SAP neurons. a** Ex-vivo setup of an isolated intact adult zebrafish heart from the Tg(*elav3:EGFP*) line allows whole-cell patch-clamp recordings of SAP neurons. The dotted line indicates a recorded neuron. **b** The SAP neurons display distinct firing pattern properties and responses to hyperpolarizing and depolarizing current step injections. The depolarization current step injections from rheobase increased by 10% of rheobase. **c** Differential representation of the neuronal types in the SAP region of the adult zebrafish heart. **d** Normalized mean values (min to max) of the electrical properties observed for the SAP neuron types that are detailed described in Supplementary Fig. 4. Every column indicates a neuron and every row represents an electrical property. **e** Sample traces showing the variable spontaneous activity of the different SAP neuron firing classes recorded in the adult zebrafish heart. Analysis of the presence (proportion) and frequency of spontaneous action potentials between the SAP neurons. **f** Representative voltage-clamp traces show the spontaneous excitatory synaptic currents (EPSCs) between the different SAP neurons. Analysis of the detected EPSC frequency and amplitude. **g** The principal component analysis (PCA) plot depicts three clusters of SAP neurons based on physiological properties (dataset presented in **d** and **e**). Cells and data are colored by the assigned cell type (as in **b**). AHP, after-hyperpolarization potential; CDF, cumulative distribution frequency; DIC, Differential interference contrast; EPSC, excitatory postsynaptic current; GFP, green fluorescent protein; *Rinput*, input resistance; RMP, resting membrane potential; SAP, sinoatrial plexus. Data are presented as mean ± SEM and as violin plots. *$P < 0.05$; **$P < 0.01$; ***$P < 0.001$; ****$P < 0.0001$; ns, not significant. For detailed statistics, see Supplementary Table 1. Source data are provided as a Source Data file.

nervous system of the heart. Collectively, our data argue against the notion that the intracardiac nervous system is a simple parasympathetic postganglionic structure that relays central efferent information[2,3,5–7]. In support of this, our single-cell sequencing, anatomical, and physiological datasets revealed several different neuronal types, including sympathetic and parasympathetic neurons and sensory neurons with an apparent neurochemical and functional diversity.

In the mammalian heart, the sinoatrial node hosts the cells responsible for the main pacemaker currents[46–49]. Similarly, in zebrafish, a simpler homolog structure to the sinoatrial node also exists[50] in the venous pole, with cells co-expressing the transcription factor Isl1 and the pacemaker cells ion channel HCN4[17,51]. Accordingly, Isl1 mutant zebrafish display cardiac defects in initiating heart contraction and regular pauses and display a high mortality rate, indicating a faulty

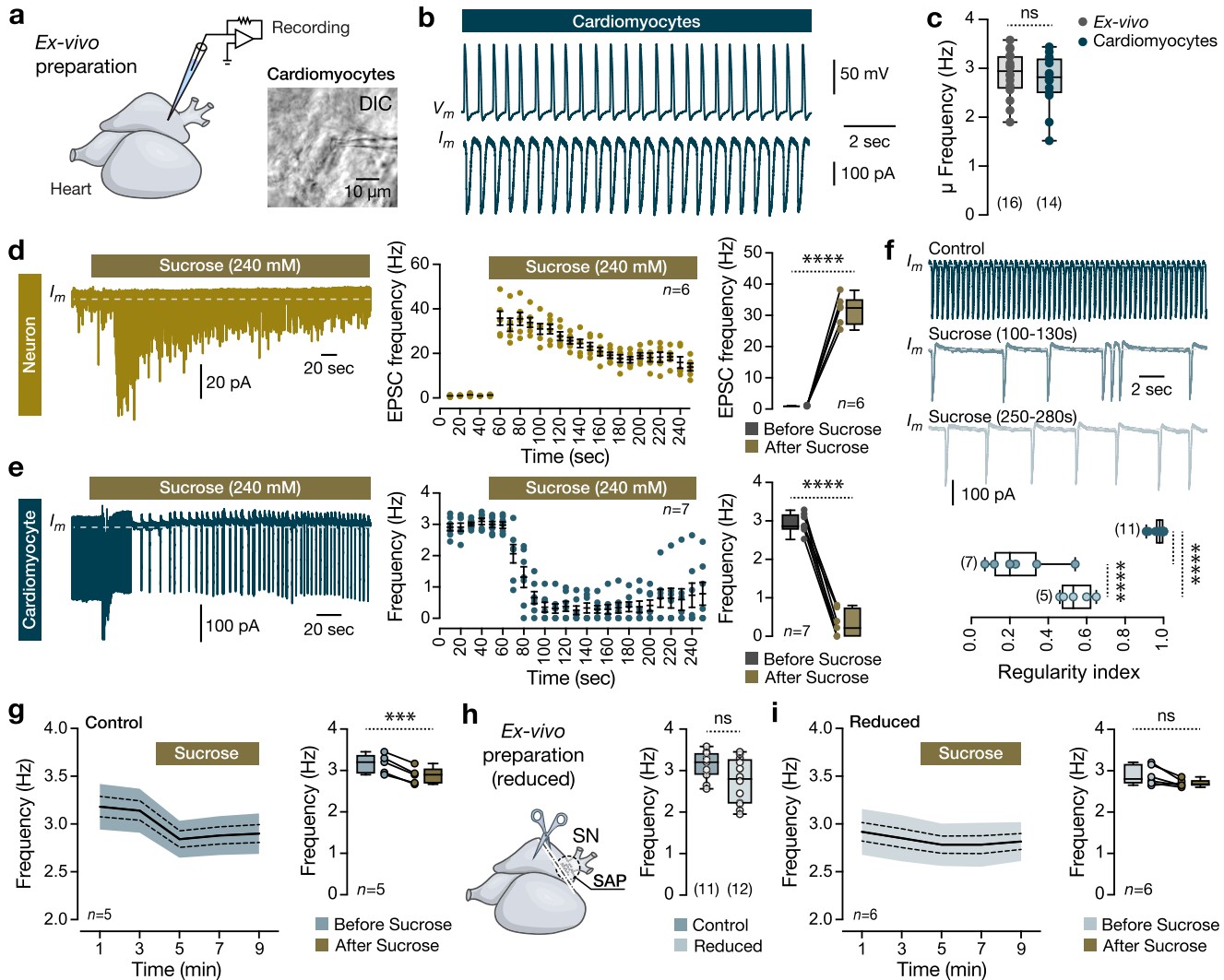

**Fig. 5 | Functional changes in heart rate. a** Ex-vivo preparation of an isolated adult zebrafish heart allows whole-cell patch-clamp recordings of atrial cardiomyocytes. Image of a recorded cardiomyocyte. **b** Representative current- and voltage-clamp traces from recorded cardiomyocytes. **c** Mean heart rate frequency (Hz) of the recorded cardiomyocytes and the ex-vivo video-recorded hearts. **d** Voltage-clamp recording from an IcNS neuron shows the increase of the detected excitatory post-synaptic currents (EPSCs) after bath application of sucrose. Data are presented as mean values ± SEM. Analysis of the EPSCs frequency (Hz) before and following the Sucrose. **e** A sample trace from a cardiomyocyte recording in voltage-clamp mode shows a disruption of the firing frequency following sucrose application. Data are presented as mean values ± SEM. Quantification of the cardiomyocyte firing before and after the application of Sucrose. **f** Sample traces from a cardiomyocyte before and after the application of sucrose and analysis of the firing regularity. **g** Time-course analysis of the heart rate frequency extracted from video recordings of the

ex-vivo adult zebrafish heart and quantification of the heart frequency before and after the application of sucrose. The shaded area defines the standard deviation, and the dotted line is the standard error of the mean (solid line). **h** Reduced ex-vivo adult heart preparation after removal of the sinus venosus (SN) and SAP neurons. Analysis of the heart average frequency (Hz). **i** Time-course analysis and quantification of heart frequency extracted from ex-vivo reduced zebrafish heart videos before and after sucrose application. The shaded area defines the standard deviation, and the dotted line is the standard error of the mean (solid line). All cells were held at −75mV (voltage-clamp recordings). DIC, Differential interference contrast; SAP, sinoatrial plexus; SN, sinus venosus. Data are presented as box plots showing the median with 25/75 percentile (box and line) and minimum–maximum (whiskers). ***$P < 0.001$; ****$P < 0.0001$; ns, not significant. For detailed statistics, see Supplementary Table 1. Source data are provided as a Source Data file.

pacemaker activity[50,52]. The pacemaker cells of the sinoatrial node are a distinct type of cell that share many similarities with both cardiomyocytes and neurons[53]. However, as the Isl1-expressing cells in the zebrafish heart are found not to express neuronal markers but to co-localized with *myl7*+ cells (Myosin Light Chain 7; a marker of cardiomyocytes), suggests that the zebrafish sinoatrial node pacemaker cells have a non-neuronal nature[17,50].

It is well known that the cardiac nodes are regulated directly by the sympathetic and parasympathetic branches of the autonomic nervous system, which relies on cholinergic and adrenergic neurotransmission[54,55]. Yet, our sequencing data indicate the expression of several additional neurotransmitter receptors, including genes

encoding glutamatergic, GABAergic, and serotonergic receptors, not only in the intracardiac neuronal cluster but also in non-neuronal cells, such as the cardiomyocytes. Accordingly, our anatomical studies have identified the presence of glutamatergic, GABAergic, and serotonergic neurons among the typical cholinergic and catecholaminergic IcNS (intracardiac nervous system) neurons, complementing earlier research conducted in zebrafish[17,36]. In addition, previous studies in rodents have shown the presence of IcNS glutamatergic vGlut2 (vesicular glutamate transporter 2) neurons, but GABAergic or serotonergic neurons were not detected[31]. This suggests that the zebrafish IcNS is neurochemically more complex than the mammalian one. As the mammalian nodal pacemaker cells expressed high levels of

glutamatergic receptors[56,57], this observation raised the question of who provides the glutamatergic input. While it has been suggested that glial cells can mediate the glutamatergic signaling to the sinoatrial node[57], our data broaden this perspective to include the IcNS glutamatergic neurons. It is important to note that the characterization of the neurotransmitter phenotype serves as a robust predictor of neuronal functionality, allowing us to infer potential roles and interactions within the intracardiac neuronal network. However, these predictions need to be substantiated through further functional studies. Hence, detailed electrophysiological, pharmacological, and molecular investigations will be essential to fully understand the complex interactions and regulatory mechanisms orchestrated by these neurotransmitters within the intracardiac neuronal network to gain a complete understanding of how this network contributes to the overall control and regulation of heart rhythm and function.

In our study, we observed that the majority of the HuC/D⁺ neurons were present in the SAP (sinoatrial plexus) region of the heart, aligning with previous findings in teleost fish[17,18]. However, we detected a smaller population of neurons in young adult animals compared to the numbers reported in older zebrafish[17], suggesting age-related dynamic variations in the organization of the intracardiac nervous system[58]. Moreover, we have identified significant biochemical and functional heterogeneity of adult zebrafish intracardiac neurons (IcNS) by revealing the presence of several types of excitatory (Glutamatergic), inhibitory (GABAergic), and modulatory (Cholinergic, Catecholaminergic, and Serotonergic) neurons. In addition, a previous electrophysiological characterization of the intracardiac neurons suggested the importance of the intracardiac neurons in heart rhythmicity[39]. Our findings are in line with this previous study[39] demonstrating the presence of bursting neurons that exhibit various rythmogenic/pacemaker physiological properties, *inter alia* SAG potential, spontaneous activity, and afterhyperpolarization rebound firing[41,42]. These intrinsic neuronal pacemaker properties could enable neuronal networks to transform unpatterned tonic input into rhythmic activity with a precise frequency range. It is thus probable that the IcNS contains a network that resembles a central pattern generator (CPG). Central pattern generators are autonomous, highly localized neuronal networks capable of generating and controlling stereotyped, rhythmic motor outputs[41,59,60]. In the vertebrate central nervous system, distinct CPGs are involved in several complex behaviors like locomotion, respiration, mastication, deglutition, urination, and ejaculation[59–64]. Yet, these networks are also found in the periphery within the enteric nervous system, which generates motility patterns linked with intestinal secretion[65,66]. While more studies are needed to confirm the presence of a CPG-like network in vertebrate hearts, in some invertebrates, it is already established that leech heart patterning activity relies on a central pattern generator network[67]. These findings establish the functional diversity of the intracardiac neurons in zebrafish, highlighting the potential complexity of their role in cardiac function and regulation.

Based on our data, it is conceivable, indeed probable, that the vertebrate heart has a two-layer localized peacemaking/control system, a nodal pacemaker, which exhibits spontaneous depolarization and activity and is responsible for generating the inherent heart rate[50,68], and a neuronal regulatory module that dictates the operational range after integrating and computing central and peripheral information. Hence, our study supports the proposed model of the heart's "little brain" by which a diverse neuronal population forms a local network that integrates and processes essential information for heart functionality[8–10].

While the zebrafish presents a prototypic vertebrate heart with a single atrium and ventricle, it is an emerging and robust model system for human-related cardiovascular studies due to the strong similarities in heart rate, cardiomyocyte action potential duration, and morphology with respect to humans[14,15]. As such, neuronal maps like those presented here, which describe distinct structural, biochemical, and functional features, provide essential guidance for future studies on the heart's nervous system's development and function. Thus, we anticipate that cell-type-specific classifications of intracardiac neurons will facilitate further functional analyses of the diverse but stereotypic neuron populations that generate and gate sensory and motor functions to control the vertebrate heart.

## Methods

### Experimental animals and Inclusion statement

Young adult (8–12 weeks old; Total Length > 1.8-2.0 cm; Standard Length > 1.55 cm) wild-type zebrafish (*Danio rerio*; AB/Tübingen, RRID: ZIRC_ZL1) and transgenic *Tg(elavl3:EGFP; vGluT1:GFP; vGluT2a:DsRed; Gad1b:GFP)* lines where used in all anatomical, and physiology experiments. All zebrafish were similar in each experiment to minimize variability due to size/age. No randomization or blinding was used to assign animals to groups. All zebrafish research conducted in this study adhered to strict ethical standards and inclusion principles to ensure the welfare and humane treatment of the animals and the scientific integrity of the research. The experimental protocols were approved by the Swedish Regional (Stockholm) Ethical Committee (ethical permit no. 13913-2022) and were implemented under the EU Directive for the care and use of laboratory animals (2010/63/EU) as well as directed according to the ARRIVE guidelines. For single-cell sequencing experiments, *Tg(elavl3:EGFP)* (n = 50; 5-month-old and 11-month-old) and *Tg(nbt:DsRed)* (n = 10; 22-month-old) lines from both sexes were used. All animal experiments followed the applicable regulations and were approved by the Institutional Animal Care and Use Committee (IACUC) at Columbia University (protocol number AC-AABN3554). All efforts were made to use the minimum number of zebrafish necessary to obtain reliable scientific data for statistical analysis. Inclusion criteria for selecting zebrafish included both male and female animals to ensure a balanced representation and to account for any sex-specific differences in the experimental outcomes. Fish of comparable age and developmental stage were used to maintain consistency and reliability in the data collected.

### Immunohistochemistry

Animals were deeply anesthetized with 0.3-0.5% tricaine methane sulfonate (MS-222). The hearts were extracted and fixed in 4% PFA at 4 °C for 2–14 h. Immunolabeling was performed in whole-mount hearts using modified protocols similar to the ones described previously for the zebrafish central nervous system[69–71]. The tissue was washed 3 times for 5 min each in PBS. Nonspecific protein binding sites were blocked with 4% normal donkey serum (NDS; D9663, Sigma-Aldrich) with 1% bovine serum albumin (BSA; A2153, Sigma-Aldrich) and 1% Triton X-100 (T8787, Sigma-Aldrich) in PBS for 1 h at room temperature (RT). Primary antibodies (see Supplementary Table 2) were diluted in 1% of the blocking solution and applied for 1–3 days at 4 °C. After thorough rinses with PBS, the hearts were incubated with the appropriate fluorescent secondary antibodies or with biotinylated secondary antibodies in 1% Triton X-100 (T8787, Sigma-Aldrich) in PBS overnight at 4 °C. When a biotinylated secondary antibody was used, streptavidin conjugated to Alexa (see Supplementary Table 2) was applied overnight at 4 °C. Finally, the tissue was thoroughly rinsed in PBS and coverslipped with 80% glycerol (G9012, Sigma-Aldrich) in PBS mounting solution.

### Single-cell transcriptomics

Single-cell transcriptomics on adult zebrafish hearts was performed as two independent replicate experiments (n = 60 hearts). Hearts were dissected and pooled together for tissue dissociation using the Neural Tissue Dissociation Kit (Miltenyi)[72,73]. Following dissociation, cells underwent filtration using a 40 μM cell strainer into a solution comprising 10 mL of 2% BSA in PBS. Centrifugation at 300 g for 10 minutes

was followed by resuspension in a cell suspension buffer containing 2% BSA in PBS, a protocol that described previously[72,73]. This cell suspension was used to sort viable GFP-positive and dsRED-positive cells as singlets using an appropriate gating strategy (Supplementary Fig. 7, Supplementary Dataset 1) together with viability indicator dyes Sytox Blue (Invitrogen, Cat No. S34857) and Dycle Ruby (Invitrogen, Cat. No. V10309) by FACS. The single-cell suspension was promptly loaded onto the 10X Chromium system[74]. The 10X libraries were sequenced via Illumina NovaSeq 6000[72,73,75,76]. In total, 9997 cells were sequenced.

## Electrophysiology

Transgenic Tg(*elavl3:EGFP*) animals were anesthetized in ice-cold extracellular solution (135.2 mM NaCl, 2.9 mM KCl, 2.1 mM $CaCl_2$, 10 mM HEPES, 10 mM glucose; pH 7.4) and fixed ventral part upwards on a surgery bed while perfused with extracellular solution. A longitudinal incision was made along the thoracic cavity and abdomen to expose and carefully remove the pericardium (a sack that encloses the heart). The hepatic portal vein, common cardinal veins, and subclavian veins were cautiously severed in order to preserve the integrity of the sinus venosus. Finally, the ventral aorta and brachial arteries were also severed, and the heart was removed and placed in a recording chamber and continuously perfused with extracellular solution. Vaseline was used to secure the heart in place, allowing access to the sinus venosus area where the SAP neurons are located. SAP neurons expressing EGFP were visually identified and their XYZ-coordinates were marked (position) using a motorized microscope equipped with a CCD camera at 60x magnification. The heartbeat was stopped by adding the photosensitive myosin II inhibitor (±)-blebbistatin (60 μM; Sigma-Aldrich, 203392) to the perfusion solution (in darkness). The contractile activity of the heart was abolished 20-30 minutes after, and whole-cell patch-clamp intracellular recordings were performed, both in voltage- and current-clamp modes. Electrodes with a resistance of 7−9 MΩ were pulled from borosilicate glass capillaries (outer diameter: 1.5 mm; inner diameter: 0.87 mm; Hilgenberg) by using a micropipette puller (model P-97, Sutter Instruments), and filled with intracellular solution (120 mM K-gluconate, 5 mM KCl, 10 mM HEPES, 4 mM $Mg_2ATP$, 0.3 mM $Na_4GTP$, 10 mM Na-phosphocreatine; pH 7.4). Electrodes were penetrated mechanically to the tissue and advanced to targeted neurons or cardiomyocytes using a motorized micromanipulator (Luigs & Neumann) applying steady positive pressure. No chemicals were used to expose or to aid the penetration of the electrodes to the tissue. The recorded signal was amplified with a Multi-Clamp 700B intracellular amplifier (Molecular Devices). Neurons and cardiomyocytes were clamped at −75 mV throughout all voltage-clamp recordings. All experiments were performed at RT (-23 °C). In some experiments, a 240 mM Sucrose solution (Sigma-Aldrich, S0389) was added and diluted to the extracellular solution to achieve the desired concentration. *Post hoc* evaluation of the recorded neurons that expressed GFP (*elav3*+) was performed after the recordings as light neutralized the effect of blebbistatin.

## Video recordings

The heart was isolated and mounted in a recording chamber described above (*see* "Electrophysiology" section) and placed under a stereo microscope (Leica Microsystem SD9, Leica) equipped with a digital camera (Leica MC170 HD). In a subset of experiments, the sinus venosus (SN) and SAP area were surgically dissected from the preparation before the mounting procedure. Video recordings (duration:10 minutes) were made, and the heartbeat frequency was evaluated at five time points (1, 3, 5, 7, and 9 minutes respectively). Drugs were delivered locally onto the neurons in the SAP/sinus venosus area using a microneedle held in a close position by a micromanipulator. Drug administration took place at minute 4 of the recording. Effects were measured at subsequent time points. A

solution of sucrose (240-300 mM; Sigma-Aldrich, S0389) was applied locally near the SAP region.

## Analysis

The raw single-cell sequencing underwent analysis using the default settings of the Cell Ranger Single Cell Software Suite (10X Genomics, version 6.1.2). On average, 97.45% of the total 763.7 million gene reads were aligned to the zebrafish genome release GRCz11 (release 105). The matrices obtained were used as input for downstream data analysis by Seurat[77]. The Seurat objects were generated by excluding cells expressing fewer than 200 genes and genes expressed in fewer than 3 cells. After filtering out the low-quality cells, Seurat objects were normalized, and for further analyses, the top 2000 variable genes were used. Upon identifying the anchors by using the *FindIntegrationAnchors* function, we integrated the two datasets (IntegrateData). The integrated Seurat object included 9508 cells with 23,118 genes. The data were scaled using all genes, and 32 PCAs (RunPCA) were identified. Cell clustering, marker gene analyses, and differential gene expression were performed using Seurat V4.1.3[73,78,79]. To visualize the gene expressions as Violin and Feature plots, we used *ggplot2* of R visualization tool. The clusters were identified using a resolution of 0.5. In total, 22 clusters were identified. The main cell types were identified by using *tnnc1a, tnnt2a*, and *myh10* for cardiomyocytes; *cdh5* and *ndufa4l2a* for endothelia cells; *krt4* and *krt15* for epicardia; *hbaa1* and *ba1* for erythroid cells; *cd74a, ptprc*, and *mpeg1.1* for immune cells; *elavl3* and *sv2a* for neural cells; *mbpa* and *mpz* for Schwann cells; *apln* and *myh11a* for vascular cells. To find differentially expressed genes, we used the *FindMarkers* function of Seurat with a 0.25 logfc threshold. GO/KEGG analysis was performed by using DEGs. For iterative clustering of neural cells, we used a subset function to create Seurat objects, including only neurons and Schwann cells (for clusters 17 and 21). We normalized and scaled the data and used *FindNeighbours* with 30 dims. Clustering was performed using resolution 1.5. We identified four subclusters. For the detection of average and percentage expression of selected genes, we used *AverageExpression* and *PrctCellExpringGene* functions, respectively.

All immunodetections of whole-mount images of the adult zebrafish heart preparations were acquired using an LSM700 or LSM800 laser scanning confocal microscope (Zeiss) using either a dry 20x or an oil-immersed 40x objective. Each examined whole-mount heart was placed accordingly to reveal the SAP area and scanned, generating a z-stack with a z-step size of 2-3 μm. Most of the presented whole-mount images were generated using a subset of the original z-stacks to enhance the visualization of our presented data. Colocalizations were detected by visual identification of structures whose color reflects the combined contribution of two antibodies or markers in the merged image. Whole-mount heart preparations are used to identify the relative position of intracardiac SAP neurons distributing up to 80 μm from the epicenter of the sinoatrial valve using Fiji/ImageJ. All morphological features, such as soma area (soma size, $\mu m^2$) and the number of cells, were measured using Fiji/ImageJ. To improve data visualization, the fluorescent images were inverted, presented as a single channel, or converted to magenta-green for color-blind readers. Digital modifications of the images (brightness and contrast) were kept to a minimum to avoid distorting biological information. Most voltage-clamp presented traces were low-pass filtered (Gaussian, 11-21 coefficients) using Clampfit (version 11.0; Molecular Devices). The EPSC amplitude was calculated as the difference between the baseline and the event's peak. All electrophysiology parameters were measured and analyzed using Clampfit (version 11.0; Molecular Devices). The RMP was determined as an average baseline activity during a 3 sec recording in gap-free mode. The statistical analyses of EPSC frequency and cardiomyocyte firing frequency in electrophysiological recordings before and after sucrose application were generated as an average frequency of 0 to 40 sec (before sucrose) and 100 to 140 sec (after

sucrose) from the same cell. For the video recordings using the ex-vivo heart preparation, statistical analyses of average frequency were performed before the application of sucrose (average frequency from 0 to 3 min) and after (average frequency from 5 to 8 min) and compared using paired Student's *t*-test. All figures and graphs were prepared using Adobe Illustrator (Adobe Systems Inc., San Jose, CA, USA; RRID: SCR_010279) and Prism (GraphPad Software Inc.; RRID: SCR_002798).

### Statistics and reproducibility

To determine the significance of differences between experimental data group means, we used parametric tests such as the two-tailed unpaired or paired Student's *t*-test and one-way ANOVA (ordinary). We followed up with *post hoc* Tukey's test to compare all groups or Dunnett's multiple comparison tests to compare groups to the control group. Statistics were performed using Prism (GraphPad Software Inc.). In all graphs presented, the significance level is denoted as $*P < 0.05$, $**P < 0.01$, $***P < 0.001$, and $****P < 0.0001$. "n.s." stands for no significance. Analyzed data are presented as mean ± s.e.m. (standard error of the mean), as violin plots, and as box plots showing the median, 25th, and 75th percentile (box and line), and minimal and maximal values (whiskers). Finally, the number of validated animals per group, cells, or events is presented as *n* values and can be found in Supplementary Table 1. For GO terms, a hypergeometric test or Fisher's exact test was applied to determine the statistical significance of the enrichment. *P* values from these tests were also adjusted for multiple comparisons using FDR correction (FDR < 0.05). To prevent any observer, selection, and confirmation biases, we conducted independent experiments and analyses multiple times, ranging from 2 to 7 times, with different investigators. In all cases, the individuals arrived at the same conclusions.

### Reporting summary

Further information on research design is available in the Nature Portfolio Reporting Summary linked to this article.

## Data availability

All data used for the analyses presented in this study are provided in the Source Data file. Detailed analyses of the data presented in this study are included in Supplementary Table 1. The single-cell sequencing dataset generated in this study has been deposited in the NCBI's GEO database under accession code GSE261619. The single-cell sequencing dataset can be accessed at NCBI's GEO with the accession number GSE261619. Further information and reasonable requests for resources and reagents should be directed to and will be fulfilled by the corresponding authors, C.K. (ck2893@cumc.columbia.edu) and K.A. (Konstantinos.ampatzis@ki.se).

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

## Acknowledgements
We thank Dr. Konstantinos Meletis for the valuable discussion of the data, Giannis Tsiverdis for technical assistance, and Ampatzis and Kizil Lab members for their discussion, comments, contributions to the project, and assistance in preparing this manuscript. This work was supported by grants from the Dr. Margaretha Nilssons Foundation (to K.A.), Erik and Edith Fernstrom Foundation (to K.A.), StratNeuro (to K.A.), Karolinska Institutet (to K.A.), Columbia University Schaefer Research Scholar Award (to C.K.), Thompson Family Foundation Program for Accelerated Medicines Exploration in Alzheimer's Disease and Related Disorders of The Nervous System (TAME-AD) (to C.K.), and Taub Institute Grants for Emerging Research (TIGER) (to C.K.). The single-cell sequencing was performed by the Single Cell Analysis Core and Columbia Genome Center at the Sulzberger Genome Center, which was funded in part through the NIH/NCI Cancer Center Support Grant P30CA013696 and used the Genomics and High Throughput Screening Shared Resource. This publication was partly supported by the National Center for Advancing Translational Sciences, National Institutes of Health, through Grant Number UL1TR001873. We also thank Erin Bush, Izabela Krupska, and Peter Sims for their support. The content is solely the responsibility of the authors and does not necessarily represent the official views of the NIH.

## Author contributions
K.A. and C.K. conceived and initiated the project. A.P., L.DV., and YW.E.D. performed the electrophysiology experiments. A.P., I.T.V. and K.K. performed anatomy and pharmacology experiments. E.Y. and P.B. performed the scRNA sequencing and the bioinformatic analyses. All authors collectively analyzed the data, discussed the results, and prepared the figures. K.A. and C.K. wrote the manuscript with input from all authors.

## Funding

## Competing interests
The authors declare no competing interests.
