## [Transparent Peer Review file · Nature Communications]

Decoding the molecular, cellular, and functional heterogeneity of zebrafish intracardiac nervous system

Corresponding Author: Dr Konstantinos Ampatzis

Version 0:

Reviewer comments:

Reviewer #1

(Remarks to the Author)

In this study, Pedroni et al. dissected the molecular, anatomical, and functional heterogeneity of the intracardiac neuronal cell types in the adult zebrafish heart. They provided a structural and functional map of neuronal diversity. Remarkably, using an ex-vivo preparation of the adult heart, the authors revealed a novel group of intracardiac neurons with peacemaking potential. The profiles will be useful for the field and hypothesis generation. However, some analyses are premature and need further development, as detailed below.

1. The molecular heterogeneity revealed by scRNAseq is highly informative. However, these molecular signatures were neither validated in vivo with gene expression assays (e.g., in situ hybridization or antibody staining) nor linked to the spatial distributions of these neuron subtypes, typology, and electrical properties described in the following analyses. Also, Schwann cells were identified in the initial clustering; however, they were omitted from the following studies. Where are these cells located, and what is their molecular signature?
2. Fig. 1. Although HuC/D is more broadly expressed than NeuN, it is unknown whether HuC/D labels all intracardiac neuron somata. Anti-acetylated-tubulin staining may be combined to mark all nerves as a reference, as shown in a previous study (PMID 25711945) cited in this manuscript.
3. The scRNA-seq analysis needs to be clarified. Are they from dissociated single cells of the entire heart or FACS-sorted GFP+ or DsRed+ cells from *elavl3:eGFP* or *not:DsRed* hearts? If from the sorted cells, why are there so few neurons? Also, the QC results of the dataset should be provided as supplementary figures or tables. In line 199, it was claimed that neuronal-related cells represent less than 1.5% of the heart cells. It is unclear how this number was derived if only sorted cells were included in the scRNA-seq analysis. This number is likely overestimated.
4. The scRNA-seq analysis, particularly the cell annotation in Figure 2, is a bit superficial. It needs to be clarified how cell clusters are annotated. What are the differences between “vascular cells” and “endothelial cells”? *krt4* and *krt15* are not typical epicardial cell markers, and they may be expressed by mural cells, endothelial cells, and cardiac valve cells. Supplementary tables with annotated marker genes should be provided. The cell number and data quality may not support a meaningful deep clustering into 22 clusters, which is unnecessary. There is no further information offered by, for example, separating 7 epicardial subtypes (some of them are quite close to cardiomyocytes). Please consider a lower clustering strength.
5. The ex vivo measurement of electrical properties is a particularly exciting aspect of this study. Would it be possible to link these properties to the typology and molecular features revealed in previous figures? It would be very informative to at least map the spatial distribution of the 4 subtypes using the GFP reporter (i.e., record cell locations while measuring the electrical properties).
6. Fig. 1d. The legend needs more details. What does the x-axis denote? Although I think it indicates the SAP, AVP, VBAP, BA, and Ventricle regions, it would be helpful to label and explain them clearly. Also, please provide an explanation of cumulative distribution frequency (CDF).
7. Fig. 2h. The heatmap is less informative than the one in Fig. S2. Most genes seem to be highly expressed in subcluster 2

and equally expressed in the remaining 3 subclusters. The differences in subclusters should be emphasized in this figure, like in 2i.

8. A recent paper describes parasympathetic and sympathetic axons in the mouse heart (PMID: 37674983). The authors may consider citing it in the current study.

9. Line 292-304. Typo, the referred figure should be figure 5, not figure 4.

Reviewer #2

(Remarks to the Author)

While the overall intent of the study is good, and there are in fact some novel and interesting findings, I have several concerns on the manuscript as it is presented in terms of overall novelty/impact of the work, critical concerns with methodology, and finally how this study is framed in the context of previously reported works, both in studies using zebrafish and in the context of the overall field of neurocardiology.

ABSTRACT

Overall, the abstract is quite superficial and would greatly benefit from providing more specific details regarding the key aspects of the study (e.g., further background/context, key experimental details such as the species used, greater details about what was actually shown). Finally, as it is written the key significance of the findings is not directly stated.

INTRODUCTION

The Introduction does not provide sufficient explanation/justification as to why the zebrafish is used in this study; it is not until the very end of the paper (Lines 378-381) that the importance/relevance of the zebrafish as an experimental model for cardiac studies is mentioned. This is crucial information and should be brought forward, even in an introductory manner in this section and has been covered in recent reviews (<https://pubmed.ncbi.nlm.nih.gov/30514520/>), and specifically for intracardiac studies recent reviews such as: <https://pubmed.ncbi.nlm.nih.gov/34821702/>.

- Line 104: "...which is located on the surface of the heart..."

This is a misleading and an oversimplification of ICNS anatomy even if the intent of the authors is to only describe the zebrafish ICNS, and a grossly misleading statement, which has been countered by most every report, if also including the mammalian ICNS.

- Line 122: "...revealed a novel group of intracardiac neurons with pacemaker/rhythmogenic properties..."

This is not in fact a novel finding. In fact it was in 1999 that Smith demonstrated tonically firing intracardiac neurons (<https://doi.org/10.1152/ajpregu.1999.276.2.R455>; which is cited elsewhere in the article), along with the other nerves types demonstrated in the current study. These findings have also been demonstrated by other groups since.

RESULTS

- Lines 194-195: "...suggesting the presence of cells under development or with neurogenic potential..."

Depending on the age of the zebrafish used in this experiment (which was not clear – see the note regarding the Methods below), this may not be surprising, as it has been shown that the populations of nerves in the heart are not stable until at least 12 months post fertilisation (see Stoyek et al, 2018, Prog Biophys Mol Biol). In fact, depending on the age of zebrafish used in this part of the study, it may be that the relative proportion and distribution of subclusters may not yet represent a stable finding, and may not reflect the proportions of cells in the adult heart.

- Lines 233-234: "Collectively, our findings reveal a neurochemical complex and heterogeneous orchestration of the adult zebrafish SAP"

While some additional neurochemical aspects are revealed in the current study (e.g. GABA/glycine), much of this neurochemical and heterogeneous complexity was previously shown in Stoyek et al., 2015 (<https://doi.org/10.1002/cne.23764> ; which is cited elsewhere in the article), and Stoyek et al., 2017 (<https://doi.org/10.1016/j.autneu.2017.07.004> ; not cited in this paper).

Further, to this the authors report that they detected approximately 77 intracardiac neurons within the zebrafish heart in the current study, which represents less than half of the total number in previous reports that they have cited at any developmental age, and yet the authors make no effort to frame their findings in the context of these previous reports.

- Lines 244-245: "Using the elicited firing responses to the current injection steps, we categorize the SAP neuron into four broad firing classes"

While this may be the first time this is shown in zebrafish, as had been previously mentioned this has been demonstrated in the mammalian heart (e.g., Smith, 1999; <https://doi.org/10.1152/ajpregu.1999.276.2.R455>).

- Supplementary Figure 4:

Data in Data in this figure would be better served to be included with the primary results in the main body of the text.

- Lines 279-280: "...the SAP network may be capable of not only regulating but also generating the heart rhythm"

How is it possible that intracardiac neurons can 'generate' the heart rhythm? Are the authors suggesting that there is a neurogenic rather than myogenic component to the zebrafish heart? If this is the case, it needs to be much more strongly argued and supported. Furthermore, this would require excitation of cardiac cells by neurons, which would require neuromuscular junctions, which have not been yet been demonstrated in the heart. In fact, it is well established that

sinoatrial cells show spontaneously excitation, which drives cardiac rhythm, including in the zebrafish heart (<https://pubmed.ncbi.nlm.nih.gov/35295582/>). Neuronal activity has to date only been shown to modulate heart rate, with no support to my knowledge, provided for involvement in generating the rhythmic excitation of the heart.

- Lines 292-301

Figures are mislabelled as 4, when they should be 5.

- Lines 289-300 "...we developed a reduced version of the heart ex-vivo preparation by dissecting the sinus venosus/SAP area in which the heart rate was unaffected"

Some clarification around this point is needed. Is the implication here that this was a reduced preparation which maintained the same heart rate as the whole heart? If so, is the implication that they are not damaging SAN cells and for this reason it was a good model?

- Figure 5, Lines 303-304 "resoundingly impl[ies] that the lcNS plays a significant role in setting and modulating the cardiac rhythm by directly affecting the nodal pacemaking system"

- The findings shown in this figure, which are not novel. It has been known for more than a century that the nervous system alters heart rate by affecting pacemaker activity and has been shown previously in zebrafish.

- How/why application of high concentration sucrose affects neuronal firing, but not cardiomyocyte function, should be explained.

- That sucrose affects heart rate in the whole heart (Fig. 5g) but does not in the reduced SAP preparation (Fig. 5i) is somewhat confusing, and does not appear to be a particularly robust result. The heart rate values in the time course plots in panels Figures 5G and 5I are much lower, and do not seem to match, the average measurements in right panel. I would ask the authors to provide rationale/justification for this, as currently it is not clear how the data in the right panels was generated.

- This potential discrepancy aside, in Figure 5I there are 2 measurements that appear to show a similar reduction in heart rate to those in Figure 5G, while the other 4 seem to show less of a reduction, but those 4 begin at a lower heart rate than any of the hearts in Fig. 5I. Were these results somehow different to begin with?

- While it is shown that heart rate is not significantly different in the reduced SAP preparation compared to the whole heart, the variability of the heart rates of the reduced SAP preparations is greatly increased, so that there is no difference between the reduced SAP preparation and whole heart seems tenuous.

DISCUSSION

- Lines 317-318 "In the mammalian heart, the sinoatrial node hosts the cells responsible for the main pacemaker currents"

The reference Cingolani et al., 2018 does not seem appropriate for the statement, there are many much more appropriate recent (and less so) reviews for such a general statement about the sinoatrial node. At the same time, it would be appropriate to include references related to the function of zebrafish sinoatrial node cells (<https://pubmed.ncbi.nlm.nih.gov/35295582/> and <https://pubmed.ncbi.nlm.nih.gov/23077655/>).

- Lines 323-325 "It is still unknown whether the sinoatrial node pacemaker cells are specialized neurons or cardiomyocytes, as they share several common features (Zhou, 2021) – This statement refers to a comment on a paper that shows that "Sinoatrial node pacemaker cells share dominant biological properties with glutamatergic neurons"

That study does not demonstrate that sinoatrial node pacemaker cells may be neurons rather than cardiomyocytes, but that the two cells types share biological properties. In comparison there is a very, very large body of literature demonstrating that sinoatrial node pacemaker cells are indeed cardiomyocytes, therefore this statement is misleading and great overreach.

-Lines 340-341: "This suggests that the zebrafish lcNS is more evolutionarily complex than the mammalian one, possibly due to a more elaborative mammalian sinoatrial node."

This is a far-reaching, and perhaps misleading, statement without providing more substantiative support.

-Lines 348-350: "In our study, we have identified significant biochemical and functional heterogeneity of adult zebrafish intracardiac neurons (lcNS) by revealing the presence of several types of excitatory, inhibitory, and modulatory neurons." Given the findings reported within this study, this is a bit of a tenuous statement without further characterization and should be clarified to more accurately represent the findings of the study.

- Lines 350-351: "...our results have confirmed the presence, for the first time, of bursting neurons that exhibit various rhythmic/pacemaker physiological properties..."

How can one both "confirm" something, and demonstrate it "for the first time"? This is indeed a confirmation – as stated above, the rhythmic firing of lcNS neurons was shown previously in Smith, 1999 – and therefore this finding is not as novel as the authors suggest.

- Lines 366-268: "These findings establish the functional diversity of the intracardiac neurons in zebrafish, highlighting the complexity of their role in cardiac function and regulation"

This should be stated as "...highlighting the POTENTIAL complexity of their role in cardiac function and regulation".

- Lines 370-373: "Based on our data, it is conceivable, indeed probable, that the vertebrate heart has a hierarchic two-layer localized pacemaking/control system, a nodal pacemaker that creates the inherent heart rate, and a neuronal regulatory module that dictates the operational range after integrating and computing central and peripheral information"

As above, what is novel about this finding is unclear. It has been known for some time that heart rate is determined by the spontaneous activity of the sinoatrial node cells, which is modulated by nervous system activity. Is the novelty is the "integration" of information by the lcNS, rather than simply relaying central nervous system activity, as has been suggested

for some time but has not yet been shown conclusively? Furthermore, it is unclear what about the current findings makes this “probable”. Preceding statements (Lines 365-368) regarding the authors hypothesis of a ‘central pattern generator’ also needs to be clarified in light of this as well.

METHODS

- Lines 389-398

A wide array of zebrafish ages are used, which makes interpretation of the results difficult, and various findings possibly unrelated. Experiments using wild-type zebrafish used 8-12 week old animals (which are termed ‘Adult’, even though zebrafish have not yet reached breeding age/sexual maturity until 12 weeks, so is misleading), experiments with Tg(elavl3:eGFP) animals used 5-month and 11-month old animals (which are very different ages, yet it is unclear in which cases each were used, or if they were grouped together, which would be inappropriate), and experiments with Tg(nbt:DsRed) animals used 22-month old animals (which are in fact more ‘aged’ zebrafish). It is unclear why such a broad range of ages was used and which results relate to which age.

- Lines 446-447

- The concentration of blebbistatin reported to be used was 60 mM, unless this is a typographical error, this represents a concentration that is more 1000x greater than that reported by any other group (which all use concentration in the uM range). If a concentration that that high was truly used, blebbistatin could have secondary effects and the authors would need to report on this. Further to this, the authors report using GFP to identify intracardiac neurons for intracellular recordings, which would assumedly use a blue excitation light, which is known to inactivate blebbistatin. Could the authors comment on this?
- Electrodes for intracellular recordings had a resistance of 7-9 Mega Ohms, which is very low for such recordings, the authors need to provide an explanation/rationale for this.
- Experiments were conducted at room temperature (~23°C), yet beating rates of the explanted hearts was ~3Hz – this is much higher than the heart rate reported by others at this sub-physiological temperature (<https://pubmed.ncbi.nlm.nih.gov/24671241/>; <https://pubmed.ncbi.nlm.nih.gov/30017908/>; <https://pubmed.ncbi.nlm.nih.gov/26730947/>) and some explanation needs to be provided by the authors.

- Lines 530-532: “...we conducted independent experiments and analyses multiple times, ranging from 2 to 7 times, with different investigators....in all cases, the individuals arrived at the same conclusions”
Where is the data showing the replicates of the experiments that came to the same conclusions as the ones shown in the paper? At minimum this information should be included as raw data in the Supplemental Information.

Version 1:

Reviewer comments:

Reviewer #1

(Remarks to the Author)

The authors have addressed my major concerns. This story will be useful for the field and for hypothesis generation.

Reviewer #2

(Remarks to the Author)

Overall, while the manuscript presents some very interesting and valid findings, there unfortunately continue to be some issues that preclude me from accepting this manuscript as it is:

Abstract: I can agree with the author’s response and revision, acknowledging that my initial take on this may have been stylistic which can be subjective.

Introduction: The included revision to the use of zebrafish is much better, especially considering the target for a broader audience.

Clarification on surface versus superficial, while seemingly trivial, is appreciated as it does have different implications comparing zebrafish to mammalian hearts (again considering a broader audience).

Line 122: While the word pacemaker may not have been used explicitly in the cited study, and the existence of CPG is not at all debated, there have been reports of rhythmic ‘phase-locked’ intracardiac neurons previously.

Lines 194-195: While agreeing with the possibility of continual regeneration and neurogenesis, and in fact it is very likely that the zebrafish heart possess a cardiac ‘stem-cell’-like population that could result in changing neuronal numbers, the argument is not one of stability. If the “zebrafish’s dynamic neurogenic and proliferative capabilities underscore the plausibility of ... developmental stages within its nervous system (at adult stages)”, how do the authors then fit their work into the scope?

Related to this is the total number of the number of neurons found in the current study, Stoyek et al. 2015 found ~200 HuC/D+ (labels neuronal cell bodies) and AcT (labels axons only) and the current study reports only 77. The authors say that only 30-40% in their study were HuC/D+ and that with the inclusion of HuC/D- the numbers would match, however if we

are comparing only HuC/D+ only this does still does not make immediate sense. If corrected for age/development with (Stoyek et al. 2018, Prog Biophys Mol Biol) the numbers are closer (~150 total HuC/D+) but there still exists a discrepancy.

Lines 233-234: I appreciate both the novel GABA/Glut findings, and the efforts to put things in better context.

In terms of the number of neurons found in the current study, Stoyek et al. 2015 found ~200 HuC/D+ (labels neuronal cell bodies) and AcT (labels axons only) and the current study reports only 77. The authors say that only 30-40% in their study were HuC/D+ and that with the inclusion of HuC/D- the numbers would match, however if we are comparing only HuC/D+ only this does still does not make immediate sense. If corrected for age/development with (Stoyek et al. 2018, Prog Biophys Mol Biol) the numbers are closer (~150 total HuC/D+) but there still exists a discrepancy.

Lines 279-280 & 289-300: I appreciate the revision for clarity on the potential for a neuronal pacemaker operation and the ex vivo preparation.

Lines 348-350: My initial point here was simply that neurotransmitter content suggests function but does not demonstrate it (which I think the authors also agree) and that the discussion of these findings should be worded as such.

Lines 370-373: The authors state that "SAN cells display spontaneous activity; however, to our knowledge, these findings are in preparations where the IcNS is also present. As such, this statement is weak..". I would suggest that authors refer to Tessadori et al. 2012 (<https://doi.org/10.1371/journal.pone.0047644>), or other reports (especially those characterising the slo mo mutant) that show spontaneous depolarisation in isolated SANC from zebrafish, as well as others in mammals (<https://doi.org/10.1113/jphysiol.2003.040501>). While this does not exclude a potential neural contribution, it should be considered.

Lines 389-398: While I can agree that a precise developmental 'staging' with zebrafish can be problematic, the steps with TL are a great step towards that. However, even in zebrafish, with a gradual senescence (large body of literature on this), there is likely to be large differences at a cellular level between a 1.5—2.0cm zebrafish that is 8-12 weeks old versus one that is 22 months old. This again goes back to the comment on neuron numbers and the considerations for 'developmental stages' in terms of the nervous system.

Version 2:

Reviewer comments:

Reviewer #2

(Remarks to the Author)

The authors have presented a work that has both confirmatory and novel components and represents a critical step in the study of the intracardiac nervous system, and does well to support the zebrafish as a relevant model to do so. I greatly appreciate the open dialogue with the authors on questions and clarifications that I have raised, as well as their open approach to answering these questions and framing their work in reference to the existing scaffolding of knowledge. At this point I feel that the authors have now addressed most all of my comments and concerns, however in the final reading I have one last minor clarification question that was somewhat raised by the authors their Revision 1 comments in terms of the neuron (ICN) identification.

The authors stated that " Neurons expressing eGFP were identified and marked (position) using a microscope equipped with a CCD camera at 60x magnification. The heartbeat was stopped by adding the photosensitive myosin II inhibitor (\pm)-blebbistatin (60 μ M; Sigma-Aldrich, 203392) to the perfusion solution (in darkness). The contractile activity of the heart was abolished 20-30 minutes after, and whole cell patch-clamp intracellular recordings were performed, both in voltage- and current-clamp modes."

My two questions on this are:

1) It is known that ICN in the zebrafish embedded in the SAP tissues (myocytes and connective tissue), were any steps taken to expose the ICN, or a was a certain position (e.g., atrial lumen/SA valve leaflets exposed) found to offer better access? In either case, a brief statement of this would help future reproducibility by readers;

and the second, and more important question:

2) The authors state that ICN were marked before application of Blebbistatin. Was this a physical marker or a microscope software-based coordinates mark based on XYZ coordinates? Clarification on this is important for both reproducibility and to ensure that the authors are in fact recording an ICN and not accidentally from a SAN cell which are known to tightly surround the IcNS in zebrafish. The classification of the ICN subtypes is very important to the authors manuscript and while the SS/A/R ICN recordings show stable pre and post RMP, there is a bit of movement in the RMP of the (B)ursting type (though this could simply just be the representative images).

A final, and very minor, note at this point also is that I believe (after being recently corrected) that EGFP is now the preferred abbreviation.

Version 3:

Reviewer comments:

Reviewer #2

(Remarks to the Author)

The authors have now addressed all of my major questions and I feel that this manuscript will now be a great addition to the field and foundation for future studies.

Point-by-Point Response to Reviewers comments:

We appreciate the valuable feedback and constructive criticism given by the reviewers. We have revised the manuscript accordingly, considering all comments and suggestions, as detailed below. The Reviewers' comments are indicated in bold and italics, while our responses are in regular style in this document. Moreover, the revised manuscript file highlights all the amended text to allow the Reviewers to track our changes.

Reviewer #1 (Remarks to the Author):

In this study, Pedroni et al. dissected the molecular, anatomical, and functional heterogeneity of the intracardiac neuronal cell types in the adult zebrafish heart. They provided a structural and functional map of neuronal diversity. Remarkably, using an ex-vivo preparation of the adult heart, the authors revealed a novel group of intracardiac neurons with peacemaking potential. The profiles will be useful for the field and hypothesis generation. However, some analyses are premature and need further development, as detailed below.

Authors: We are thankful for the Reviewer's very helpful feedback and appreciation of the significance of our work. We also anticipate that this study will complement previous studies, enhancing the foundations for future more detailed dissection of the neuronal intracardiac networks, and generate new experimentally testable hypotheses.

1. The molecular heterogeneity revealed by scRNAseq is highly informative. However, these molecular signatures were neither validated in vivo with gene expression assays (e.g., in situ hybridization or antibody staining) nor linked to the spatial distributions of these neuron subtypes, typology, and electrical properties described in the following analyses. Also, Schwann cells were identified in the initial clustering; however, they were omitted from the following studies. Where are these cells located, and what is their molecular signature?

Authors: Thank you for your valuable feedback. The primary objective of this manuscript is to characterize neuronal cells and their electrophysiological properties. While we acknowledge that integrating spatial, molecular, and electrophysiological properties would provide a more comprehensive understanding, we believe that such an extensive study is beyond the scope of this publication. This level of integration would require the development of new analytical and statistical tools, as well as highly powered studies due to the limited number of neurons available per animal, which affects reliable statistical power and effect size.

We appreciate your understanding as we plan to incorporate these aspects in our future research endeavors. Additionally, in Figure 3, we have presented data addressing some of the queries. This includes the spatial, topological, and electrophysiological properties of neurons, particularly highlighted in Fig. 3j, which documents the relative distance to the center of SAP. Subsequently, we focused on nervous system-related Clusters 17 and 21 to delineate subclusters, resulting in the identification of four distinct subclusters. To clarify their characteristics, we now include Sankey diagrams in **Supplementary Figure 4**, illustrating genes demarcating different cell properties. We found that Subclusters 1 and 4 represent neuroepithelial cells, Subcluster 2 neurons, and Subcluster 3 Schwann cells. Moreover, we identified differential neurotransmitter-related gene expressions in Subcluster 2, justifying the spatial and functional characterization of neuronal types as shown in main Fig. 3. Although Schwann cells were not a primary focus of this manuscript, we have added **Supplementary Table 3**, which details the gene expression signatures of the Schwann cell subcluster (Subcluster 3).

To address the spatial localization of different subclusters, we performed triple immunostaining for three markers of subclusters: acetylated tubulin (for neurons, subcluster 2), myelin basic protein (Schwann cells for subcluster 3, and Cadherin 1 for subclusters 1 and 4, neuroepithelia). We provide this information in **Figure 1 - for Reviewers** (please see below). We are working on this aspect for another study and will use this image as preliminary results; however, if the reviewer is willing to include this staining in the manuscript, we can integrate this into Figure 2 of the main manuscript. To further validate the scSeq data in Subcluster 2, we performed SV2 and HuC/D stainings (**Figure 2 – for reviewers**).

We hope this additional data provides clarity and supports the comprehensiveness of our study within its defined scope.

Figure 1 - for Reviewers. Immunolabeling for myelin basic protein (red), acetylated tubulin (green), and Cadherin 1 (white) with DAPI counterstain in adult zebrafish heart. Lower panels show colocalization scores within 5-micron resolution. Myelin basic protein and acetylated tubulin-positive cells are distinct but in close proximity, while Cadherin⁺ cells can be found in distinct locations compared to other cells. These results support our findings that these cell types are distinct and spatially organized.

Figure 2 - for Reviewers. Immunodetection of HuC/D and SV2 verifying the expression of the *elav3*, *eval4*, and *sv2a* genes in the neuronal population in the adult zebrafish heart.

2. Fig. 1. Although HuC/D is more broadly expressed than NeuN, it is unknown whether HuC/D labels all intracardiac neuron somata. Anti-acetylated-tubulin staining may be combined to mark all nerves as a reference, as shown in a previous study (PMID 25711945) cited in this manuscript.

Authors: Thank you for this comment. HuC/D and acetylated tubulin marks different regions of the neurons as

HuC/D is predominantly labeling the perinuclear region in the cell. Indeed, the prior study conducted by Stoyek et al. (2015) offered a meticulous mapping of the intracardiac nervous system of zebrafish, utilizing a combination (cocktail) of the HuC/D antibody and acetylated-tubulin antibody. In our current research, we aimed to complement this previous work by specifically focusing on the HuC/D⁺ neurons by using HuC/D immunolabeling but also HuC/D-driven GFP in transgenic animals. We also provided Zn12 and HuC/D co-stainings such as in Figure 1. HuC/D⁺ cells were almost entirely associated with Zn12 filaments, suggesting that the majority of the neurons are HuC/D positive. Therefore, we considered our decision to characterize HuC/D neurons was safe. We now provide the acetylated tubulin staining in our previous response as **Figure 1 – for reviewers**.

Our decision to focus on HuC/D was driven by several crucial considerations:

Avoiding redundancy: Our intention was to avoid redundancy and repetition of previously well-reported findings and data. By narrowing our focus to HuC/D⁺ neurons, we aimed to contribute novel anatomical and functional insights without duplicating existing research efforts.

Neurochemical characterization: Our neurochemical characterization, as illustrated in Figure 3, revealed a significant aspect: the absence of glutamatergic, cholinergic, GABAergic, serotonergic, or catecholaminergic neurons that were HuC/D⁻. This observation is pivotal for systems neuroscience studies, where understanding the organization and functionality of neural networks heavily relies on the classical small neurotransmitters and their receptors. By specifically examining HuC/D⁺ neurons, we sought to provide foundational knowledge essential for unraveling the intricacies of the intracardiac nervous system.

Facilitating future studies: Importantly, our study aims to establish a framework and resources to facilitate future investigations into the intracardiac nervous system. By focusing on HuC/D⁺ neurons, we aim to provide a robust foundation for employing advanced methodologies such as optogenetics, chemogenetics, and dual electrophysiological recordings. These techniques, in conjunction with our findings, will enable researchers to delve deeper into the functional properties and regulatory mechanisms of the intracardiac neural circuitry.

Transgenic reporter lines: Leveraging transgenic reporter lines for HuC and D positive neurons, our study offers a unique opportunity to target and record from these neurons specifically. These reporter lines not only enhance the precision of our research but also serve as an invaluable resource for developing further intersectional genetic tools. By employing techniques such as optogenetics and chemogenetics, researchers can manipulate and control HuC/D⁺ neurons with unprecedented specificity, thereby advancing our understanding of their roles in cardiac function and regulation.

References:

Stoyek MR, Croll RP, Smith FM. Intrinsic and extrinsic innervation of the heart in zebrafish (*Danio rerio*). *J Comp Neurol*. 2015 Aug 1;523(11):1683-700. Epub 2015 Apr 9. PMID: 25711945.

3. The scRNA-seq analysis needs to be clarified. Are they from dissociated single cells of the entire heart or FACS-sorted GFP⁺ or DsRed⁺ cells from *elavl3:eGFP* or not: *DsRed* hearts? If from the sorted cells, why are there so few neurons? Also, the QC results of the dataset should be provided as supplementary figures or tables. In line 199, it was claimed that neuronal-related cells represent less than 1.5% of the heart cells. It is unclear how this number was derived if only sorted cells were included in the scRNA-seq analysis. This number is likely overestimated.

Authors: Thank you for your insightful comment. The scRNA-seq analysis presented in this study was conducted on dissociated cells from the entire heart. Given the very low percentage of neuronal cells in the heart, it was not feasible to sort exclusively for neurons from this tissue and subsequently perform single-cell RNA sequencing. Several studies, including a recent *Danio* heart atlas (<https://www.mpi-hlr.de/danioheartatlas>), did not obtain a specific neuronal cluster. The logistic challenges of dissecting, dissociating, and sorting a sufficient number of cells from approximately 400 animals in a single day made it impractical to limit the analysis to GFP⁺ or DsRed⁺ neurons alone.

To ensure an adequate number of cells for robust scRNA-seq analysis, we expanded the gating of fluorescence to include both GFP⁺ and DsRed⁺ cells as well as non-fluorescent cells. This approach, detailed in

Supplementary Figure 7, allowed us to capture a broader range of cell types, including cardiomyocytes and other non-neuronal cells. As a result, while our gating strategy enriched for neuronal cells, it also included a significant contribution from other cell types. To clarify our methodology and support our findings, we have added a new **Supplementary Data 1**, which includes the gating strategy and the respective percentages of the different cell types captured. This gating strategy inevitably led to an estimated representation of neuronal cell types at approximately 1.5% of the total heart cells. We acknowledge the reviewer's concern that this estimate might be overestimated, given the heterogeneous nature of the heart tissue and the potential for contribution from other cell types. Yet, our fluorescent sorting percentages were around this range.

Additionally, we have included a new **Supplementary Figure 2**, which provides the quality control (QC) results of our single-cell RNA sequencing analyses. This figure includes metrics such as the distribution of unique molecular identifiers (UMIs) and the percentage of mitochondrial gene expression across cells, ensuring transparency in our data quality. Moreover, we have added **Supplementary Table 4**, which is a comprehensive Excel file detailing the differentially expressed genes in neural cell clusters compared to other cell types. This supplementary data aims to clarify the gene expression profiles of neuronal cells within our dataset.

We hope these additions will enhance the clarity of our methodology and address the reviewer's concerns regarding the estimation and representation of neuronal cells in our scRNA-seq analysis.

4. The scRNA-seq analysis, particularly the cell annotation in Figure 2, is a bit superficial. It needs to be clarified how cell clusters are annotated. What are the differences between “vascular cells” and “endothelial cells”? krt4 and krt15 are not typical epicardial cell markers, and they may be expressed by mural cells, endothelial cells, and cardiac valve cells. Supplementary tables with annotated marker genes should be provided. The cell number and data quality may not support a meaningful deep clustering into 22 clusters, which is unnecessary. There is no further information offered by, for example, separating 7 epicardial subtypes (some of them are quite close to cardiomyocytes). Please consider a lower clustering strength.

Authors: Thank you for your constructive feedback. We appreciate the opportunity to clarify our scRNA-seq analysis and the cell annotations presented in Figure 2.

Our study performed cell clustering using a resolution of 0.5, which we determined to be the optimal balance for capturing the diversity of cell types in our dataset without over-fragmenting the clusters. We acknowledge the reviewer's concern about the depth of clustering and agree that a more granular approach could risk losing significant information about the neural cell clusters, which are relatively low in abundance. Therefore, the 22 clusters presented represent the minimum number required to provide meaningful insights based on the power of our current dataset.

We acknowledge the reviewer's point regarding the markers krt4 and krt15. These markers were included to provide a broad characterization of cells within the epicardial lineage but may indeed be expressed in a variety of other cell types, including mural and cardiac valve cells. To address this, we have updated our annotations and provided additional marker genes that are more specific to annotated cell cluster cells in **Figure 2b** with a new dot plot.

Regarding the clustering strength, while reducing the resolution further may simplify the clustering, it risks losing critical information, particularly for neural cell clusters. This is a trade-off we had to consider carefully, given the current size and quality of our dataset. In future studies, we plan to expand our sequencing efforts to include a larger number of cells, which will allow for a more refined clustering analysis and potentially result in different cluster delineations.

We did the following updates: **Figure 2c** and **Figure 2d**: We have updated these figures to reflect the new clustering results and annotations based on the reviewer's feedback. The revised figures now provide a clearer representation of the identified cell types and their respective markers.

We hope these clarifications and updates address the reviewer's concerns and enhance the robustness of our scRNA-seq analysis.

5. The ex vivo measurement of electrical properties is a particularly exciting aspect of this study. Would

it be possible to link these properties to the typology and molecular features revealed in previous figures? It would be very informative to at least map the spatial distribution of the 4 subtypes using the GFP reporter (i.e., record cell locations while measuring the electrical properties).

Authors: Thank you for highlighting the significance of our *ex vivo* measurements of electrical properties. We agree that linking these properties to the typology and molecular features identified in previous figures would provide valuable insights. However, we would like to clarify that in our study, we focused on neuronal subtypes, particularly those in Subcluster 2. The other subclusters consist of Schwann cells (Subcluster 3) and neuroepithelial cells (Subclusters 1 and 4), which fall outside the scope of our current analysis, although we determined that they probably are involved in the functional unit organization of neurons as per spatial localization determined in **Figure 1 - for Reviewers** (please see above comment #1). The primary aim of this manuscript is to characterize the electrical properties of neurons, which are comprehensively presented in Fig. 3, showing the topology and spatial distribution of the neuronal cells in Subcluster 2.

Nevertheless, to address the comments as much as possible, we performed the following updates to **Fig. 2j**:

- Schwann cells (Subcluster 3): While Schwann cells were not the focus of our electrical property measurements, we have performed myelin basic protein (MBP) staining to illustrate their distribution (Reviewer Figure 1).

- Neuroepithelia (Subclusters 1 and 4): These cells were identified using Cadherin staining (**Figure 1 - for Reviewers**), but as they are not the focus of our current manuscript, detailed mapping of their spatial distribution and electrical properties falls beyond the scope of this study.

We hope that these additional details and clarifications help to address the reviewer's comments and enhance the overall understanding of our findings within the context of this manuscript.

6. Fig. 1d. The legend needs more details. What does the x-axis denote? Although I think it indicates the SAP, AVP, VBAP, BA, and Ventricle regions, it would be helpful to label and explain them clearly. Also, please provide an explanation of cumulative distribution frequency (CDF).

Authors: Indeed, the x-axis indicates the different regions of the heart based on a color-coded approach that is a common practice to avoid redundancy, and we used the same style throughout the whole manuscript. In Figure 1d, the color coding is the same as in Figure 1c, which is an issue as the color-coding key is not provided for this panel for both the size and the cumulative distribution frequency. We have now modified the panels to accommodate this information.

Cumulative frequency analysis is the analysis of the frequency of occurrence of values and is calculated by adding each frequency from a frequency distribution table to the sum of its predecessors. The power of cumulative frequency distribution is used to determine the number of observations that differentiate between different data sets (in this case, the size of the neurons). As such, the CDF can be used not only to reveal the different proportions of neurons of a particular size but also to make predictions if needed. Thus, it constitutes a very useful presentation tool.

7. Fig. 2h. The heatmap is less informative than the one in Fig. S2. Most genes seem to be highly expressed in subcluster 2 and equally expressed in the remaining 3 subclusters. The differences in subclusters should be emphasized in this figure, like in 2i.

Authors: Thank you for your comment regarding the heatmaps in Figure 2h and Supplementary Figure 2. We appreciate your observations and would like to clarify the distinct purposes these heatmaps serve within our study.

The heatmap in Supplementary Figure 2 (new **Supplementary Figure 3**) is designed to be an unbiased representation, displaying the top expressed markers for each respective subcluster. This approach is customary for delineating the molecular signatures of the clusters and provides a comprehensive overview of the gene expression profiles across all subclusters. The top 50 gene expressions are shown distinctly per cluster. In contrast, the heatmap presented in Figure 2h is a custom selection focused on neuron and neurotransmitter-related genes. These genes were manually selected to highlight specific neurotransmitter responsiveness and potential neuronal signatures within Subcluster 2. This custom heatmap is tailored to

emphasize the functional aspects of the neuronal subcluster, making it particularly informative for understanding the unique characteristics and functions of these neurons in the context of our study.

We understand the need to clearly differentiate the roles and content of these heatmaps. To address this, we have updated the legend in Supplementary Figure 2 (new **Supplementary Figure 3**) to explicitly state that it represents an unbiased selection of top expressed markers for each subcluster. This clarification should help distinguish the heatmap's purpose from that of Figure 2h, which is focused on specific neuronal and neurotransmitter-related genes.

Both heatmaps contribute valuable information to our narrative by offering complementary perspectives on the data. The unbiased heatmap in Supplementary Figure 2 provides a broad view of the molecular landscape, while the custom heatmap in Figure 2h focuses on the neuronal features that are central to our analysis. We also added Sankey diagrams (**Supplementary Figure 4**) that show the distribution of key gene expressions for different subclusters and, hence, cell types. We hope these clarifications enhance the understanding of the different heatmaps and their significance in our study.

8. A recent paper describes parasympathetic and sympathetic axons in the mouse heart (PMID: 37674983). The authors may consider citing it in the current study.

Authors: It is now included in our references.

9. Line 292-304. Typo, the referred figure should be figure 5, not figure 4.

Authors: We apologize for the typographical error and appreciate the Reviewer for bringing it to our attention. It has been rectified in the updated version of the manuscript.

Reviewer #2 (Remarks to the Author):

While the overall intent of the study is good, and there are in fact some novel and interesting findings, I have several concerns on the manuscript as it is presented in terms of overall novelty/impact of the work, critical concerns with methodology, and finally how this study is framed in the context of previously reported works, both in studies using zebrafish and in the context of the overall field of neurocardiology.

Authors: We greatly appreciate the Reviewer's comprehensive and in-depth assessment of our work, as well as the numerous constructive comments aimed at substantially improving it. Our current study was largely inspired and motivated by the seminal studies by Dr. Smith, Dr. Armour, Dr. Chatelier, Dr. Adams, and others, who highlighted the significance of studying the vertebrate intracardiac nervous system (IcNS), a crucial yet often overlooked neuronal network. Our goal was to build upon these foundational findings and complement them with additional resources, information, and data to further engage the scientific community in IcNS research.

We firmly consider that the IcNS is of significant importance not only for understanding cardiac functionality and its translational implications but also for studying neuronal plasticity, heart regeneration, and neurodegenerative disorders frequently associated with heart conditions.

Based on the overall comments we received from Reviewer #2, we would like to point out that this manuscript serves a **dual purpose**: it is both a research article and a resource article. As a resource article, it aims to provide a substantial informational database and tools that are broadly useful and engaging to the scientific community, potentially opening new avenues for experimental systems neuroscience investigations to address critical neurobiological questions. Simultaneously, it proposes new biological insights derived from analyzing the datasets, which we aimed to achieve in this manuscript. Therefore, the style of the manuscript in terms of data presentation, writing, and abstract was designed to fulfill this dual purpose.

We hope this clarifies our approach and the manuscript's intended contribution to the field.

ABSTRACT

Overall, the abstract is quite superficial and would greatly benefit from providing more specific details regarding the key aspects of the study (e.g., further background/context, key experimental details such as the species used, greater details about what was actually shown). Finally, as it is written the key significance of the findings is not directly stated.

Authors: We acknowledge the Reviewer's concern regarding the abstract's format, which is non-classical, as we usually see in specialized journals. The primary objective of our study is to present the various types of IcNS neurons using a diverse array of methods. Consequently, we aim to make this research accessible and comprehensible to a broad audience, extending beyond specialized fields. This goal informed our decision to select an interdisciplinary journal. As such, the abstract complies with the journal's guidelines, limitations, and style, ensuring alignment with the Journal's policies.

In response to the Reviewer's feedback, we have revised the abstract to include some additional pertinent information.

INTRODUCTION

The Introduction does not provide sufficient explanation/justification as to why the zebrafish is used in this study; it is not until the very end of the paper (Lines 378-381) that the importance/relevance of the zebrafish as an experimental model for cardiac studies is mentioned. This is crucial information and should be brought forward, even in an introductory manner in this section and has been covered in recent reviews (<https://pubmed.ncbi.nlm.nih.gov/30514520/>), and specifically for intracardiac studies recent reviews such as: <https://pubmed.ncbi.nlm.nih.gov/34821702/>.

Authors: We understand the reviewer's point; thus, we have revised the manuscript to include more information regarding using the zebrafish model in the Introduction section. In our view, animal models are important tools that help us address major biological questions. We apologize if we did not convey this principle clearly to the Reviewer, as our primary focus is neuroscience, not specifically zebrafish biology. There are numerous

examples where zebrafish provided a discovery knowledge, which was later validated in mammalian models and humans. We anticipate that our study will also serve for novel information for the field.

- Line 104: "...which is located on the surface of the heart..." This is a misleading and an oversimplification of ICNS anatomy even if the intent of the authors is to only describe the zebrafish ICNS, and a grossly misleading statement, which has been countered by most every report, if also including the mammalian ICNS.

Authors: Thank you for this comment. We revised the sentence to avoid a potential semantic confusion. We welcome specific suggestions on how to phrase this localization. We changed the sentence to "...which is embedded within the superficial layers of the heart wall..."

We interpreted the misleading nature of the following sentence from the reviewer as: "...located on the heart's surface...". We were motivated to use the term "surface" based on existing literature. Several previous studies stated:

"...located on the posterior **surfaces** of the atria..." (Giannino et al., 2024)

"...ganglia and interconnecting nerves on the posterior **surface** of the left atrium..." (Ashton et al., 2020)

"...the ganglionic neurons and surrounding satellite glial cells form a flat sheet on the **surface** of the underlying atrial muscle tissue..." (Harper and Adams, 2021)

"...the most numerous of the 5-HT-LIR cell types within the atrium was associated with the luminal **surfaces** of the atrium..." (Stoyek et al., 2017)

"...GFP+ structures are visible at the outer **surface** of the myocardium." (Tessadori et al., 2012)

"...These anatomical studies have elucidated the location of GP, the majority of them are located in supraventricular regions, either on the epicardium or embedded in fat pads on the **surface** of the heart hilum.." (Fedele and Brand, 2020 referred to Wake and Brack, 2016)

"...was mainly located at the epicardial **surface**." (Lizot et al., 2022)

In addition, in a previous study using goldfish and cryosections, the authors observed the location of the neurons in the superficial subepicardium (Newton et al., 2014).

If the reviewer's argument is based on the semantic difference between superficial and surface, and the Reviewer perceives that in our text, we mean that neurons somehow are floating on the surface of the heart, that was not our intention. The neurons are embedded on the surface of the heart, which means that they are located in the most superficial parts of the heart wall. Similar to the neurons of the enteric nervous system (Hao et al., 2012; Hawkins et al., 2013). We are open to further clarifications if needed.

References:

- Achanta S, Gorky J, Leung C, Moss A, Robbins S, Eisenman L, Chen J, Tappan S, Heal M, Farahani N, Huffman T, England S, Cheng ZJ, Vadigepalli R, Schwaber JS. A Comprehensive Integrated Anatomical and Molecular Atlas of Rat Intrinsic Cardiac Nervous System. *iScience*. 2020 Jun 26;23(6):101140. Epub 2020 May 26. PMID: 32460006.
- Ashton JL, Argent L, Smith JEG, Jin S, Sands GB, Smaill BH, Montgomery JM. Evidence of structural and functional plasticity occurring within the intracardiac nervous system of spontaneously hypertensive rats. *Am J Physiol Heart Circ Physiol*. 2020 Jun 1;318(6):H1387-H1400. Epub 2020 May 1. PMID: 32357112.
- Fedele, L., and Brand, T. (2020). The Intrinsic Cardiac Nervous System and Its Role in Cardiac Pacemaking and Conduction. *J Cardiovasc Dev Dis* 7.
- Giannino G, Braia V, Griffith Brookles C, Giacobbe F, D'Ascenzo F, Angelini F, Saglietto A, De Ferrari GM, Dusi V. The Intrinsic Cardiac Nervous System: From Pathophysiology to Therapeutic Implications. *Biology (Basel)*. 2024 Feb 7;13(2):105. PMID: 38392323.
- Hao MM, Lomax AE, McKeown SJ, Reid CA, Young HM, Bornstein JC. Early development of electrical excitability in the mouse enteric nervous system. *J Neurosci*. 2012 Aug 8;32(32):10949-60. PMID: 22875929.
- Hawkins EG, Dewey WL, Anitha M, Srinivasan S, Grider JR, Akbarali HI. Electrophysiological characteristics of enteric neurons isolated from the immortomouse. *Dig Dis Sci*. 2013 Jun;58(6):1516-27. Epub 2013 Jan 31. PMID: 23371009.
- Lizot, G., Pasqualin, C., Tissot, A., Pagès, S., Faivre, J.-F., and Chatelier, A. (2022). Molecular and functional characterization of the mouse intrinsic cardiac nervous system. *Heart Rhythm* 19, 1352–1362.
- Newton, C.M., Stoyek, M.R., Croll, R.P., and Smith, F.M. (2014). Regional innervation of the heart in the goldfish, *Carassius auratus*: a confocal microscopy study. *Journal of Comparative Neurology* 522, 456–478.
- Stoyek, M.R., Jonz, M.G., Smith, F.M., and Croll, R.P. (2017). Distribution and chronotropic effects of serotonin in the zebrafish heart. *Autonomic Neuroscience: Basic and Clinical* 206, 43–50.

Tessadori F, van Weerd JH, Burkhard SB, Verkerk AO, de Pater E, Boukens BJ, Vink A, Christoffels VM, Bakkens J. Identification and functional characterization of cardiac pacemaker cells in zebrafish. *PLoS One*. 2012;7(10):e47644. doi: 10.1371/journal.pone.0047644. Epub 2012 Oct 16. PMID: 23077655; PMCID: PMC3473062.

Wake E, Brack K. Characterization of the intrinsic cardiac nervous system. *Auton Neurosci*. 2016 Aug;199:3-16. doi: 10.1016/j.autneu.2016.08.006. Epub 2016 Aug 6. PMID: 27568996.

- **Line 122: "...revealed a novel group of intracardiac neurons with pacemaker/rythmogenic properties..."**

This is not in fact a novel finding. In fact it was in 1999 that Smith demonstrated tonically firing intracardiac neurons (<https://doi.org/10.1152/ajpregu.1999.276.2.R455>; which is cited elsewhere in the article), along with the other nerves types demonstrated in the current study. These findings have also been demonstrated by other groups since.

Authors: Currently, there is no evidence that tonic neuronal firing upon depolarization current step injection indicates pacemaker/rhythmogenic properties. As such, in the seminal publication suggested by the reviewer, "pacemaker" or "rythmogenic" properties were not presented, therefore this paper does not provide an intuitive basis for the previous demonstration of intracardiac neurons with pacemaker/rythmogenic properties. Thus, we respectfully believe that the novelty still lies within the context we identified these neurons. We consider that it is important to note that the typical rhythmic neuronal networks require the presence of pacemaker/rhythmogenic neurons, along with excitatory and inhibitory neurons (Grillner, 2003; Grillner, 2006; Grillner and Jessell, 2009; Goulding, 2009; Straub, 2009; Morquette et al., 2012; Kiehn, 2016). Our study is the first to report the presence of both glutamatergic and GABAergic neurons in the IcNS, further highlighting our work's novelty.

Several eminent researchers have extensively characterized pacemaker/rhythmogenic neuronal networks, known as central pattern generators (CPGs), including:

Eve Marder (invertebrate stomatogastric ganglion; Brandeis University, USA),

Sten Grillner (locomotor CPG in cats, lampreys, and rodents; Karolinska Institutet, Sweden),

Abdel El Manira (zebrafish locomotor CPG; Karolinska Institutet, Sweden),

Keith Sillar (Xenopus CPG; University of St Andrews, UK)

Jack Feldman (preBötzing Complex, breathing CPG; UCLA, USA),

Ansgar Büschges (Central pattern generating networks in insects, University of Cologne, Germany),

Ole Kiehn (mammalian CPG for locomotion; Copenhagen University, Denmark)

These investigators, among others studying various rhythmogenic systems (such as locomotion, chewing, breathing, vomiting, and ejaculation across different animal models), have demonstrated that neurons capable of generating rhythmic activity exhibit very specific properties. These properties are dependent on the different ionic currents involved in their activity, as noted in numerous studies (Bucher et al., 2015; Harris-Warrick, 2010; Marder and Bucher, 2001; Song et al., 2020). Some of the key characteristics include:

- **Burst firing pattern** (intermittent discharge of rapid action-potential sequences)
- Prominent **SAG potential**
- **After hyperpolarization rebound**
- **Continuous spontaneous activity** (without current application in a gap-free mode)

To our knowledge, no previous studies on intracardiac neurons demonstrate these properties. Thus, our finding of neurons with pacemaker properties within the intracardiac nervous system (IcNS) **is a novel finding**. However, it is important to note that the presence of pacemaker-like neurons alone does not imply that the IcNS functions as a CPG-like network. The maintenance of the oscillations does not only arise from the properties of individual neurons (such as in pacemaker cells), but also from the coupling structure of the network. Thus, further research is required to elucidate the connectivity and functional orchestration of IcNS neurons to determine if a "Heart-CPG network" exists. Our study lays the groundwork for these essential ongoing and future investigations. We kindly request the reviewer to take this context into account.

References:

Bucher, Dirk; Haspel, Gal; Golowasch, Jorge; and Nadim, Farzan (December 2015) Central Pattern Generators. In: eLS. John Wiley & Sons, Ltd: Chichester. DOI: 10.1002/9780470015902.a0000032.pub2

- Goulding M. Circuits controlling vertebrate locomotion: moving in a new direction. *Nat Rev Neurosci*. 2009 Jul;10(7):507-18. PMID: 19543221.
- Grillner S, Jessell TM. Measured motion: searching for simplicity in spinal locomotor networks. *Curr Opin Neurobiol*. 2009 Dec;19(6):572-86. Epub 2009 Nov 10. PMID: 19896834.
- Grillner S. Biological pattern generation: the cellular and computational logic of networks in motion. *Neuron*. 2006 Dec 7;52(5):751-66. PMID: 17145498.
- Grillner S. The motor infrastructure: from ion channels to neuronal networks. *Nat Rev Neurosci*. 2003 Jul;4(7):573-86. PMID: 12838332.
- Harris-Warrick RM. General principles of rhythmogenesis in central pattern generator networks. *Prog Brain Res*. 2010;187:213-22. PMID: 21111210.
- Kiehn O. Decoding the organization of spinal circuits that control locomotion. *Nat Rev Neurosci*. 2016 Apr;17(4):224-38. Epub 2016 Mar 3. PMID: 26935168.
- Marder E, Bucher D. Central pattern generators and the control of rhythmic movements. *Curr Biol*. 2001 Nov 27;11(23):R986-96. PMID: 11728329.
- Morquette P, Lavoie R, Fhima MD, Lamoureux X, Verdier D, Kolta A. Generation of the masticatory central pattern and its modulation by sensory feedback. *Prog Neurobiol*. 2012 Mar;96(3):340-55. Epub 2012 Feb 9. PMID: 22342735.
- Song J, Pallucchi I, Ausborn J, Ampatzis K, Bertuzzi M, Fontanel P, Picton LD, El Manira A. Multiple Rhythm-Generating Circuits Act in Tandem with Pacemaker Properties to Control the Start and Speed of Locomotion. *Neuron*. 2020 Mar 18;105(6):1048-1061.e4. Epub 2020 Jan 22. PMID: 31982322.
- Straub, V.A. (2009). Central Pattern Generator. In: Binder, M.D., Hirokawa, N., Windhorst, U. (eds) *Encyclopedia of Neuroscience*. Springer, Berlin, Heidelberg.

RESULTS

- Lines 194-195: “...suggesting the presence of cells under development or with neurogenic potential...”

Depending on the age of the zebrafish used in this experiment (which was not clear – see the note regarding the Methods below), this may not be surprising, as it has been shown that the populations of nerves in the heart are not stable until at least 12 months post fertilisation (see Stoyek et al, 2018, Prog Biophys Mol Biol). In fact, depending on the age of zebrafish used in this part of the study, it may be that the relative proportion and distribution of subclusters may not yet represent a stable finding, and may not reflect the proportions of cells in the adult heart.

Authors: Regarding the age of the animals, we address this issue extensively in the corresponding comment below regarding the Methods. Our single cell dataset contains cells from older animals, which should address the mentioned concern (e.g., developmental programs are very likely turned off).

The assertion regarding the supposed instability of findings concerning the presence of neuron-related cells under development warrants clarification. Across both invertebrates and vertebrates, the nervous system undergoes continuous development. Zebrafish, in particular, possess a remarkable innate ability to regenerate its nervous system, as well as various tissues and organs, including the heart, throughout its life stages. This phenomenon is well-documented in scientific literature. In our sentence, we aimed at indicating that there might be neurogenesis going on in the zebrafish heart.

Studies have consistently demonstrated neurogenesis—the production of new neurons—in the nervous system of zebrafish at all life stages (Adolf et al., 2006; Ampatzis and Dermon, 2007; Ampatzis et al., 2012; Chang et al., 2021; Ghosh and Hui, 2016; Grandel et al., 2006; Zupanc and Sîrbulescu, 2011) including the peripheral nervous system (El-Nachef and Bronner, 2020; McCallum et al., 2020). Moreover, in response to injury or other stimuli, this neurogenic capacity is further augmented (Kizil et al., 2012; Reimer et al., 2008; Chang et al., 2021; Becker and Becker, 2022). Consequently, it is plausible that the zebrafish nervous system harbors numerous newborn cells and neurons at various developmental stages, which makes the concept of stability context-dependent and relative.

The reviewer's comment pertains to single-sequencing data. In this dataset, cells are analyzed post-dissociation of the heart, with only viable cells included in subsequent analyses. Therefore, the data provide estimations of cell proportions rather than precise numerical counts. Single-cell datasets are primarily used as discovery tools that provide a high-resolution view of cellular diversity and function. This initial discovery phase is crucial for uncovering novel cell types, states, and interactions that are not apparent through bulk sequencing methods. These findings form the foundation for subsequent validation and functional studies. In our paper, we used single-cell sequencing to identify key cell types and their properties, which we then validated through additional

experimental techniques, ensuring the robustness and accuracy of our results. Thus, while single-cell datasets offer estimations of cell proportions, their primary strength lies in their ability to reveal intricate details of cellular heterogeneity and function, aligning with the primary objective of our research. Nevertheless, the data robustly identify cell types, which aligns with the primary objective of the presented research.

In our study, we chose to focus on HuC/D neurons for reasons elaborated upon in our response to comment #2 for reviewer #1. While comprehensive characterization of all cell types and developmental stages is undoubtedly valuable, such an endeavor exceeds the scope of the current study.

In conclusion, the zebrafish's dynamic neurogenic and proliferative capabilities underscore the plausibility of diverse cell populations and developmental stages within its nervous system (at adult stages). Our study's focus on HuC/D neurons aligns with the specific objectives and constraints of the dataset under consideration without detracting from the significance of other cell types and stages, which remain important subjects for future investigation.

References:

- Adolf B, Chapouton P, Lam CS, Topp S, Tannhäuser B, Strähle U, Götz M, Bally-Cuif L. Conserved and acquired features of adult neurogenesis in the zebrafish telencephalon. *Dev Biol.* 2006 Jul 1;295(1):278-93. Epub 2006 May 4. PMID: 16828638.
- Ampatzis K, Dermon CR. Sex differences in adult cell proliferation within the zebrafish (*Danio rerio*) cerebellum. *Eur J Neurosci.* 2007 Feb;25(4):1030-40. PMID: 17331199.
- Ampatzis K, Makantasi P, Dermon CR. Cell proliferation pattern in adult zebrafish forebrain is sexually dimorphic. *Neuroscience.* 2012 Dec 13;226:367-81. Epub 2012 Sep 19. PMID: 23000628.
- Becker T, Becker CG. Regenerative neurogenesis: the integration of developmental, physiological and immune signals. *Development.* 2022 Apr 15;149(8):dev199907. Epub 2022 May 3. PMID: 35502778.
- Chang W, Pedroni A, Bertuzzi M, Kizil C, Simon A, Ampatzis K. Locomotion dependent neuron-glia interactions control neurogenesis and regeneration in the adult zebrafish spinal cord. *Nat Commun.* 2021 Aug 11;12(1):4857. PMID: 34381039.
- El-Nachef WN, Bronner ME. *De novo* enteric neurogenesis in post-embryonic zebrafish from Schwann cell precursors rather than resident cell types. *Development.* 2020 Jul 13;147(13):dev186619. PMID: 32541008.
- Ghosh S, Hui SP. Regeneration of Zebrafish CNS: Adult Neurogenesis. *Neural Plast.* 2016;2016:5815439. Epub 2016 Jun 13. PMID: 27382491.
- Grandel H, Kaslin J, Ganz J, Wenzel I, Brand M. Neural stem cells and neurogenesis in the adult zebrafish brain: origin, proliferation dynamics, migration and cell fate. *Dev Biol.* 2006 Jul 1;295(1):263-77. Epub 2006 Apr 4. PMID: 16682018.
- Kizil C, Kyritsis N, Dudczig S, Kroehne V, Freudenreich D, Kaslin J, Brand M. Regenerative neurogenesis from neural progenitor cells requires injury-induced expression of Gata3. *Dev Cell.* 2012 Dec 11;23(6):1230-7. Epub 2012 Nov 15. PMID: 23168169.
- McCallum S, Obata Y, Fourli E, Boeing S, Peddie CJ, Xu Q, Horswell S, Kelsh RN, Collinson L, Wilkinson D, Pin C, Pachnis V, Heanue TA. Enteric glia as a source of neural progenitors in adult zebrafish. *Elife.* 2020 Aug 27;9:e56086. PMID: 32851974.
- Reimer MM, Sörensen I, Kuscha V, Frank RE, Liu C, Becker CG, Becker T. Motor neuron regeneration in adult zebrafish. *J Neurosci.* 2008 Aug 20;28(34):8510-6. PMID: 18716209.
- Zupanc GK, Sîrbulescu RF. Adult neurogenesis and neuronal regeneration in the central nervous system of teleost fish. *Eur J Neurosci.* 2011 Sep;34(6):917-29. PMID: 21929625.

- Lines 233-234: “Collectively, our findings reveal a neurochemical complex and heterogeneous orchestration of the adult zebrafish SAP”

While some additional neurochemical aspects are revealed in the current study (e.g. GABA/glycine), much of this neurochemical and heterogeneous complexity was previously shown in Stoyek et al., 2015 (<https://doi.org/10.1002/cne.23764> ; which is cited elsewhere in the article), and Stoyek et al., 2017 (<https://doi.org/10.1016/j.autneu.2017.07.004> ; not cited in this paper).

Authors: Indeed, we agree with the Reviewer that we did not extensively discuss our data with previous studies, especially the ones mentioned here in zebrafish. We have now revised our manuscript accordingly to include these studies as correctly commented that our study enhances and complements these previous seminal findings.

Further, to this the authors report that they detected approximately 77 intracardiac neurons within the zebrafish heart in the current study, which represents less than half of the total number in previous reports that they have cited at any developmental age, and yet the authors make no effort to frame their findings in the context of these previous reports.

Authors: Thank you for this comment. The reviewer's comment addresses the detection of approximately 77 intracardiac neurons within the zebrafish heart in our current study, noting that this number is less than half of the total reported in previous studies.

In our previous text (previous version lines 144-146) describing Figure 1e, we stated that we detected ~77 HuC/D⁺ neurons (not all neurons) in the SAP area (not in the whole heart). Since the SAP contained ~75% of the heart HuC/D⁺ neurons, the total number of HuC/D positive neurons in the heart is expected to be more than 100 neurons (HuC/D⁺). In a previous study (Stoyek et al., 2015), the authors very elegantly mapped the entire population of neurons in the intracardiac nervous system of zebrafish by combining antibodies against HuC/D and acetylated-tubulin, and they observed approximately 200 neurons in the SAP area alone. Our data are in line with this study (Stoyek et al., 2015), as in Figure 2, we found that only 30-40% of the neurons are expressing HuC/D (Subcluster 2 and part of Subcluster 3). Therefore, the number of neurons from both studies is similar, and for the SAP, our study suggests the presence of approximately 200 neurons (HuC/D⁺ and HuC/D⁻) in line with Stoyek et al., 2015.

Motivated by the comment raised here by the Reviewer, we perform additional new experiments to verify the number of neurons in the SAP area. By adding 6 more animals (as in revised Figure 2e), the number of SAP neurons slightly increased to 81 ± 3.6 , not that different from before. Here, it is important to state that we defined the SAP area as the area extending 80 μm from the epicenter of the valve, and we excluded neurons in the Sinus Venosus (SN) area. As such, it is likely the previous study used a broader definition of the SAP area and neurons that exist there, with the possibility of including at least part of the SN neurons. Accordingly, we re-analyze some of the hearts to compare the number of neurons between our defined SAP area and the SAP broader region. We found that neurons in the extended SAP region, including part of SN are 122 ± 7.5 (see **Figure 3 – for Reviewers**). Understanding the potential issue in defining the SAP area, we now state this clearly in our revised manuscript version (*analysis paragraph in the Methods section*). Therefore, we think that the observed discrepancy between studies lies in the definition of SAP area and the use of HuC/D vs. HuC/D+tubulin.

Figure 3 – for Reviewers. The number of neurons in the SAP area (as defined by the authors) compared with the extended SAP area (including part of the ventricle wall and part of SN). Unpaired *t*-test (two-tailed) $P < 0.0001$ ($t=5,417$, $df=29$).

As we previously mentioned (see comment #2 for Reviewer #1), our goal was not to replicate the experiments already published by Stoyek et al. (2015). Instead, our study aims to complement this existing work by focusing on HuC/D-positive neurons. This choice is based on our neurochemical studies, which revealed no expression of classical small-molecule neurotransmitters (acetylcholine, GABA, glutamate, serotonin, catecholamines) in any of the HuC/D-negative neurons.

This distinction is crucial as we aim to establish a foundational framework for future systems neuroscience investigations that will heavily rely on inter-neuronal connectivity. Understanding classical neurotransmission is essential for determining the connectivity patterns between different intracardiac neuronal populations and their targets. Future studies can leverage this information to perform dual (or multiple) simultaneous electrophysiological recordings and manipulate neurons with specific neurotransmitter phenotypes using advanced techniques like optogenetics or chemogenetics.

By focusing on HuC/D positive neurons and their neurotransmitter profiles, we pave the way for a deeper exploration of intracardiac neuronal networks and their functional roles. This approach not only builds upon previous findings but also sets the stage for innovative research aimed at unraveling the complexities of the intracardiac nervous system.

Additionally, it is important to clarify that single-cell sequencing (scSeq) data and analyses differ significantly from traditional cell counting methods. Single-cell sequencing is designed to capture the diversity and characteristics of individual cells rather than provide precise numerical counts. Techniques such as k-means clustering, UMAP (Uniform Manifold Approximation and Projection), and t-SNE (t-distributed Stochastic Neighbor Embedding) are employed to visualize and analyze the high-dimensional data, revealing distinct cell populations and their states.

The number of cells detected in single-cell sequencing does not directly reflect absolute cell ratios due to several factors:

1. **Cell sorting and viability:** Only viable cells are included post-dissociation, and some cell types may have higher survival rates through the sorting process, affecting the relative proportions observed despite all cell types are detected.
2. **Technical variability:** Methods such as gel integration and hanging drop sequencing can influence cell recovery rates and representation, potentially leading to discrepancies in cell numbers compared to traditional methods.
3. **Experimental parameters:** Variations in experimental parameters, including sequencing depth and coverage, can impact the detection and representation of certain cell types.

Our study also focuses on identifying and characterizing intracardiac neurons through scSeq, which provides valuable insights into their properties and interactions. The observed number of neurons, therefore, serves as an estimation rather than an exact count, consistent with the primary objective of scSeq to reveal cellular heterogeneity and function. Importantly, we conducted several subsequent validations in our paper to confirm the findings from our single-cell analyses. These validations included additional experimental techniques to ensure the robustness and accuracy of our results, framing our findings within the context of previous reports and further elucidating the roles of these neurons within the zebrafish heart.

References:

Stoyek MR, Croll RP, Smith FM. Intrinsic and extrinsic innervation of the heart in zebrafish (*Danio rerio*). *J Comp Neurol*. 2015 Aug 1;523(11):1683-700. Epub 2015 Apr 9. PMID: 25711945.

- Lines 244-245: “Using the elicited firing responses to the current injection steps, we categorize the SAP neuron into four broad firing classes”.

While this may be the first time this is shown in zebrafish, as had been previously mentioned this has been demonstrated in the mammalian heart (e.g., Smith, 1999; <https://doi.org/10.1152/ajpregu.1999.276.2.R455>).

Authors: Indeed, we are not the first that we tried to characterize the electrophysiological properties of the intracardiac neurons. As such, at the beginning of the section, we acknowledge several investigators have worked with the same subject before, and we cited several of these studies: “*The study of the electrophysiological properties of the vertebrate IcNS neurons using isolated in-vitro preparations (Selyanko, 1992; Edwards et al., 195; Smith, 1999; Sato et al., 2020; Lizot et al., 2022)*...” Among these studies is also the one indicated by the Reviewer.

Regarding our statement, “Using the elicited firing responses to the current injection steps, we categorize the SAP neuron into four broad firing classes,” we would like to emphasize that this reflects our experimental data, that we categorize them initially into 4 types. Previous studies identified 2 or 3 electrical distinct types of intracardiac neurons. Yet, non-previous studies reported the presence of the bursting neurons and the display of their pacemaker properties.

- Supplementary Figure 4:

Data in Data in this figure would be better served to be included with the primary results in the main body of the text.

Authors: We agree that there is some redundancy in the presentation of these data. We present our findings in these two ways because they reveal different information. The data in the supplementary Figure 4 are made in the conventional way, where each electrical property is investigated independently for the different neuron populations, which is better for revealing statistical differences in the data pool. Yet, this approach is extremely biased as the electrical properties of the neurons are all linked together. Figure 4d reveals the relative pattern

of changes between different neural types; it allows the readers to understand the relative differences for each individual neuron (a single neuron per column). This is an unbiased presentation, and this dataset is used to generate the PCA plot in Figure 4g. Therefore, we consider that the current presentation is unbiased and crucial for increasing the readability of our work, and we ask the permission of the Reviewer to keep this presentation of our work.

- Lines 279-280: "...the SAP network may be capable of not only regulating but also generating the heart rhythm"

How is it possible that intracardiac neurons can 'generate' the heart rhythm? Are the authors suggesting that there is a neurogenic rather than myogenic component to the zebrafish heart? If this is the case, it needs to be much more strongly argued and supported. Furthermore, this would require excitation of cardiac cells by neurons, which would require neuromuscular junctions, which have not been yet been demonstrated in the heart. In fact, it is well established that sinoatrial cells show spontaneously excitation, which drives cardiac rhythm, including in the zebrafish heart (<https://pubmed.ncbi.nlm.nih.gov/35295582/>). Neuronal activity has to date only been shown to modulate heart rate, with no support to my knowledge, provided for involvement in generating the rhythmic excitation of the heart.

Authors: Up to today, the field perceives the intracardiac nervous system contribution as solely a modulatory system, and our data here also support this role. However, the presence of a small population of neurons (Bursting) with pacemaker/rythmogenic properties broadens this view and generates a valid possibility that the lcNS could also be able to generate its own rhythm.

As we also discussed in our manuscript, the heart may have a two-layer pacemaking system, one neuronal and one non-neuronal. While we understand the position of the Reviewer, there are many demonstrations that different rythmogenic networks act together (simultaneously) and can influence each other, in some cases even directing the functionality of one to the other to achieve a desirable functional output (Powell et al., 2021; Song et al., 2020). While this is a possibility that we are obliged to bring to the field based on our findings, further studies are needed to resolve the validity of this hypothesis. Thus, we state in our text that this may be a possible option, but we do not assertively claim that this is the case.

Regarding the comment of the Reviewer that such a system will require neuromuscular junctions that have not yet been reported in the heart using the conventional idea of skeletal neuro-musculature, we respectfully do not share this view for the following reasons:

1. If the Reviewer agrees that intracardiac nervous system (lcNS) neurons modulate the heart, it is important to consider the mechanisms by which this modulation occurs. The most plausible explanation involves the release of neurotransmitters from these neurons, which then bind to and influence cardiomyocytes and/or sinoatrial node cells. This can happen either through volume transmission release or by forming non-conventional neuromuscular synaptic contacts. Our studies provide support for this mechanism, as we have observed the expression of several types of neurotransmitter receptors in non-neuronal cells, including cardiomyocytes and other heart cells (in main Figure 2e). This receptor expression suggests that heart cells are responsive to neurotransmitter signals, reinforcing the notion that neurotransmitter release from lcNS neurons plays a critical role in modulating heart function. This evidence aligns with the concept that neurotransmitter-mediated communication is fundamental to the functional dynamics between lcNS neurons and cardiac cells, providing a plausible pathway for the observed modulatory effects.

2. In addition to chemical synapses, neurons can also communicate directly through gap junction channels, not only between them (Song et al., 2016; Pedroni et al., 2024) but also with glia (Frões et al., 1999; Alvarez-Maubecin et al., 2000; Spray et al., 2019) but also with muscles. In *Xenopus*, for instance, during development, the communication between neurons and muscles relies on gap junctions (Allen and Warner, 1991).

While it is currently unclear what form neurons use to communicate and influence the cardiomyocytes and sinoatrial node cells, our data, in line with many previous studies, support the view that communication must be present.

References:

- Powell DJ, Marder E, Nusbaum MP. Perturbation-specific responses by two neural circuits generating similar activity patterns. *Curr Biol.* 2021 Nov 8;31(21):4831-4838.e4. Epub 2021 Sep 9. PMID: 34506730.
- Song J, Pallucchi I, Ausborn J, Ampatzis K, Bertuzzi M, Fontanel P, Picton LD, El Manira A. Multiple Rhythm-Generating Circuits Act in Tandem with Pacemaker Properties to Control the Start and Speed of Locomotion. *Neuron.* 2020 Mar 18;105(6):1048-1061.e4. Epub 2020 Jan 22. PMID: 31982322.
- Alvarez-Maubecin V, Garcia-Hernandez F, Williams JT, Van Bockstaele EJ. Functional coupling between neurons and glia. *J Neurosci.* 2000 Jun 1;20(11):4091-8. PMID: 10818144.
- Song J, Ampatzis K, Björnfors ER, El Manira A. Motor neurons control locomotor circuit function retrogradely via gap junctions. *Nature.* 2016 Jan 21;529(7586):399-402. Epub 2016 Jan 13. PMID: 26760208.
- Pedroni A, Dai YE, Lafouasse L, Chang W, Srivastava I, Del Vecchio L, Ampatzis K. Neuroprotective gap-junction-mediated bystander transformations in the adult zebrafish spinal cord after injury. *Nat Commun.* 2024 May 21;15(1):4331. PMID: 38773121.
- Spray DC, Iglesias R, Shraer N, Suadican SO, Belzer V, Hanstein R, Hanani M. Gap junction mediated signaling between satellite glia and neurons in trigeminal ganglia. *Glia.* 2019 May;67(5):791-801. Epub 2019 Feb 4. PMID: 30715764.
- Fróes MM, Correia AH, Garcia-Abreu J, Spray DC, Campos de Carvalho AC, Neto MV. Gap-junctional coupling between neurons and astrocytes in primary central nervous system cultures. *Proc Natl Acad Sci U S A.* 1999 Jun 22;96(13):7541-6. PMID: 10377451.
- Allen F, Warner A. Gap junctional communication during neuromuscular junction formation. *Neuron.* 1991 Jan;6(1):101-11. PMID: 1986772.

- Lines 292-301

Figures are mislabelled as 4, when they should be 5.

Authors: We sincerely apologize for the typographical error and greatly appreciate the Reviewer for bringing it to our attention. We have promptly rectified it in the updated version of the manuscript.

- Lines 289-300 “...we developed a reduced version of the heart ex-vivo preparation by dissecting the sinus venosus/SAP area in which the heart rate was unaffected” Some clarification around this point is needed. Is the implication here that this was a reduced preparation which maintained the same heart rate as the whole heart? If so, is the implication that they are not damaging SAN cells and for this reason it was a good model?

Authors: No. This is a misunderstanding regarding our reduced heart preparation. We apologize for the way we conveyed the message here. Developing and using the reduced *ex-vivo* heart preparation model, we aim to remove most of the SAP HuC/D neurons. The heart rate was found to be reduced, possibly due to subsequent damage also to SAN cells; however, among the different trials, this reduction was not statistically significant. Even if we had a statistically significant lower heart rate, this does not affect the outcome of our experiments. The experiments here show that removing a large proportion of HuC/D⁺ neurons, thus a large number of neuronal terminals, and applying Sucrose solution did not alter the heart rate. That, in turn, means that IcNS neuron release is necessary and sufficient to affect heart rhythmicity. This experimental approach represents an entry point to support that the release of neurotransmitters can affect the operation of the heart. Having this in mind we now use optogenetic tools to selectively manipulate specific neuronal types to reveal the importance of the different neurons and transmitters in heart functionality. Furthermore, current ongoing studies using laser ablation of the HuC/D neurons or neurochemical ablation by neurotoxins yield similar findings. We anticipate that we will soon be able to dissect the particular IcNS neuron types that can differentially affect (increase or decrease) the heart rhythm.

- Figure 5, Lines 303-304 “resoundingly impl[ies] that the IcNS plays a significant role in setting and modulating the cardiac rhythm by directly affecting the nodal pacemaking system”

- The findings shown in this figure, which are not novel. It has been known for more than a century that the nervous system alters heart rate by affecting pacemaker activity and has been shown previously in zebrafish.

Authors: Indeed, the significance of the nervous system in modulating and influencing heart rhythm has been well-established and recognized for many decades. However, we have not addressed this aspect in our current study. Our focus was not on the role and importance of the extracardiac nervous system (sympathetic and parasympathetic). Instead, we concentrated on the intrinsic cardiac nervous system (IcNS) in our preparation. Given that the heart is decentralized, our experiments support previous findings that have directly or indirectly

suggested the IcNS modulates the sinoatrial node (SAN). The experiments here provide additional confirmation and proof of concept.

It is a common practice in scientific research to support and complement ideas and findings with alternative approaches. The novelty of our work lies not only in the overall outcome but also in the alternative methods and tools used to pave the way for gaining deeper insights into the IcNS system.

- How/why application of high concentration sucrose affects neuronal firing, but not cardiomyocyte function, should be explained.

Authors: The application of sucrose solution is a well-established method in neurophysiology for triggering the immediate release of the readily releasable pool (RRP) of synaptic vesicles without exciting the neurons (Hubbard et al., 1968; Rosenmund et al., 1996; Kaeser and Regehr, 2017). This method operates independently of action potentials and initially does not influence any neuronal firing. In our study, we utilized this approach to reveal the net effect of various neurotransmitters released by the intracardiac nervous system (IcNS) on heart activity by recording the activity of cardiomyocytes. This experimental design allowed us to demonstrate that neurotransmitter release from IcNS neurons collectively impacts cardiomyocytes' activity frequency and regularity. This experiment provides clear evidence that the cumulative release of neurotransmitters significantly influences cardiomyocyte behavior, thereby supporting the notion that IcNS neurons play a crucial role in modulating cardiac function through a form of neuron-cardiomyocyte communication.

References:

Hubbard JI, Jones SF, Landau EM. An examination of the effects of osmotic pressure changes upon transmitter release from mammalian motor nerve terminals. *J Physiol.* 1968. 197:639-57. doi: 10.1113/jphysiol.1968.sp008579.
 Kaeser PS, Regehr WG. The readily releasable pool of synaptic vesicles. *Curr Opin Neurobiol.* 2017. 43:63-70. doi: 10.1016/j.conb.2016.12.012.
 Rosenmund C, Stevens CF. Definition of the readily releasable pool of vesicles at hippocampal synapses. *Neuron.* 1996. 16:1197-207. doi: 10.1016/s0896-6273(00)80146-4.

- That sucrose affects heart rate in the whole heart (Fig. 5g) but does not in the reduced SAP preparation (Fig. 5i) is somewhat confusing, and does not appear to be a particularly robust result. The heart rate values in the time course plots in panels Figures 5G and 5I are much lower, and do not seem to match, the average measurements in right panel. I would ask the authors to provide rationale/justification for this, as currently it is not clear how the data in the right panels was generated.

Authors: We hope that, based on our previous responses regarding the reduced preparation (comments above) and the function of sucrose (explained just above), we can convey the outcomes of these experiments. Experiment-to-experiment and animal-to-animal variability account for minor changes in frequency. Additionally, reduced preparations involved removing parts of the heart, and despite our best efforts to maintain consistency across all hearts, slight experimental variations are inevitable.

In our reduced preparations, we observed lower frequencies, which we have addressed and plotted in Figure 5h. These variations did not affect the overall experimental outcome. Crucially, our results demonstrate that removing most HuC/D neurons and their terminals does not significantly alter heart frequency following sucrose application. The small discrepancies in frequencies shown in Figures 5h and 5i are due to differences in sample size: Figure 5h includes data from 12 animals, while the sucrose experiments in Figure 5i were conducted on 6 animals. Variability based on which 6 of the 12 animals were used can result in slight deviations, but this does not affect the experimental conclusion in Figure 5i, where a paired *t*-test compared the same heart before and after sucrose application.

We also apologize for previously omitting the method of analysis used in Figures 5g and 5i from the manuscript. In these figures, data points represent the average frequency before (from 0 to 3 minutes) and after sucrose application (from 5 to 8 minutes), and comparisons were made using a paired *t*-test. We have now revised the manuscript to include this information in the methods section.

- This potential discrepancy aside, in Figure 5I there are 2 measurements that appear to show a similar reduction in heart rate to those in Figure 5G, while the other 4 seem to show less of a reduction, but

those 4 begin at a lower heart rate than any of the hearts in Fig. 5I. Were these results somehow different to begin with?

Authors: We consider that we covered this concern in our response above.

- While it is shown that heart rate is not significantly different in the reduced SAP preparation compared to the whole heart, the variability of the heart rates of the reduced SAP preparations is greatly increased, so that there is no difference between the reduced SAP preparation and whole heart seems tenuous.

Authors: We acknowledge that heart frequency is trending downward in reduced preparations. Yet, the statistical analyses showed that this reduction is not significant. As in all scientific studies, we also relied on statistical analyses to describe and present our data and the presence or absence of differences. Still, we plotted all the individual data points in all figures to allow readers to observe even the not statistically significant tendencies. Throughout the manuscript, all data are described based on the outcomes derived after statistical analyses and detailed presented in Supplementary Table 1. We would like to express our preference towards trusting rigorous statistical analysis over subjective visual assessments.

DISCUSSION

- Lines 317-318 “In the mammalian heart, the sinoatrial node hosts the cells responsible for the main pacemaker currents”

The reference Cingolani et al., 2018 does not seem appropriate for the statement, there are many much more appropriate recent (and less so) reviews for such a general statement about the sinoatrial node. At the same time, it would be appropriate to include references related to the function of zebrafish sinoatrial node cells (<https://pubmed.ncbi.nlm.nih.gov/35295582/> and <https://pubmed.ncbi.nlm.nih.gov/23077655/>).

Authors: We followed the Reviewer's suggestion.

- Lines 323-325 “It is still unknown whether the sinoatrial node pacemaker cells are specialized neurons or cardiomyocytes, as they share several common features (Zhou, 2021) – This statement refers to a comment on a paper that shows that “Sinoatrial node pacemaker cells share dominant biological properties with glutamatergic neurons”

That study does not demonstrate that sinoatrial node pacemaker cells may be neurons rather than cardiomyocytes, but that the two cells types share biological properties. In comparison there is a very, very large body of literature demonstrating that sinoatrial node pacemaker cells are indeed cardiomyocytes, therefore this statement is misleading and great overreach.

Authors: We appreciate and agree with the Reviewer's perspective. In alignment with much of the literature, as accurately noted by the Reviewer, we acknowledge that sinoatrial node (SAN) cells are likely cardiomyocytes, at least in zebrafish. However, we must also recognize the significant similarities between SAN cells and neurons. These similarities include ion channels, transcription factors, and marker genes, as well as the absence of principal cardiomyocyte structures such as myofibrils and T-tubules (Protze et al., 2017; Satoh, 2003). Additionally, several sequencing studies, including very recent ones, have identified neuronal-like cells and classified them as SAN cells (Farah et al., 2024).

Our intention was to present the unique nature of SAN cells. Nevertheless, our text aligns with the Reviewer's view. It is crucial to emphasize and support studies on the intracardiac nervous system (IcNS) because its significance is often overlooked by both the neuroscience and cardiac research communities. As such, we revised the corresponding text.

References:

- Farah EN, Hu RK, Kern C, Zhang Q, Lu TY, Ma Q, Tran S, Zhang B, Carlin D, Monell A, Blair AP, Wang Z, Eschbach J, Li B, Destici E, Ren B, Evans SM, Chen S, Zhu Q, Chi NC. Spatially organized cellular communities form the developing human heart. *Nature*. 2024 Mar;627(8005):854-864. Epub 2024 Mar 13. PMID: 38480880.
- Protze SI, Liu J, Nussinovitch U, Ohana L, Backx PH, Gepstein L, Keller GM. Sinoatrial node cardiomyocytes derived from human pluripotent cells function as a biological pacemaker. *Nat Biotechnol*. 2017 Jan;35(1):56-68. Epub 2016 Dec 12. PMID: 27941801.

Sato H. Sino-atrial nodal cells of mammalian hearts: ionic currents and gene expression of pacemaker ionic channels. *J Smooth Muscle Res.* 2003 Oct;39(5):175-93. PMID: 14695028.

-Lines 340-341: “This suggests that the zebrafish IcNS is more evolutionarily complex than the mammalian one, possibly due to a more elaborative mammalian sinoatrial node.” This is a far-reaching, and perhaps misleading, statement without providing more substantiative support.

Authors: We agree that this statement is inaccurate, as we do not have evidence. We now revised it to tune it down.

-Lines 348-350: “In our study, we have identified significant biochemical and functional heterogeneity of adult zebrafish intracardiac neurons (IcNS) by revealing the presence of several types of excitatory, inhibitory, and modulatory neurons.”

Given the findings reported within this study, this is a bit of a tenuous statement without further characterization and should be clarified to more accurately represent the findings of the study.

Authors: We are puzzled by the assertion that our statement is tenuous. Our study clearly demonstrates the presence of neurochemically distinct neurons in the intracardiac nervous system (IcNS). It is well-established in neuroscience that certain neurotransmitters have specific functions: Glutamate typically acts as an excitatory neurotransmitter, while GABA serves as an inhibitory neurotransmitter. Other neurotransmitters, such as Acetylcholine, Catecholamines, and Serotonin, can have diverse roles depending on the receptors present on the post-synaptic cells and often function as neuromodulators. Neuromodulators interact with primary neurotransmitters to enhance excitatory or inhibitory responses, influencing synaptic activity in ways that are often long-lasting and context-dependent. We believe that this fundamental knowledge in neuroscience underpins the validity of our findings.

Our single-cell sequencing data further corroborate the biochemical and functional heterogeneity of these neurons. Single-cell RNA sequencing allowed us to identify distinct clusters of neurons based on their gene expression profiles, revealing the presence of multiple neuron types with different neurotransmitter signatures. These findings provide a high-resolution map of the cellular diversity within the IcNS, supporting our statement about the heterogeneity of these neurons.

We acknowledge that non-neuroscientists might find this information tenuous. Therefore, in response to the Reviewer's comment, we have revised the specific sentence in the manuscript to improve clarity for a broader audience.

- Lines 350-351: “...our results have confirmed the presence, for the first time, of bursting neurons that exhibit various rhythmic/pacemaker physiological properties...”

How can one both “confirm” something, and demonstrate it “for the first time”? This is indeed a confirmation – as stated above, the rhythmic firing of IcNS neurons was shown previously in Smith, 1999 – and therefore this finding is not as novel as the authors suggest.

Authors: We respectfully disagree with the reviewer and explained in the above comments elsewhere why our finding is novel. We change the semantics by removing the “confirmation” statement.

- Lines 366-268: “These findings establish the functional diversity of the intracardiac neurons in zebrafish, highlighting the complexity of their role in cardiac function and regulation” This should be stated as “...highlighting the POTENTIAL complexity of their role in cardiac function and regulation”.

Authors: We revised the text as suggested.

- Lines 370-373: “Based on our data, it is conceivable, indeed probable, that the vertebrate heart has a hierarchic two-layer localized pacemaking/control system, a nodal pacemaker that creates the inherent heart rate, and a neuronal regulatory module that dictates the operational range after integrating and computing central and peripheral information”

As above, what is novel about this finding is unclear. It has been known for some time that heart rate is determined by the spontaneous activity of the sinoatrial node cells, which is modulated by nervous system activity. Is the novelty is the “integration” of information by the IcNS, rather than simply relaying central nervous system activity, as has been suggested for some time but has not yet been shown conclusively? Furthermore, it is unclear what about the current findings makes this “probable”. Preceding statements (Lines 365-368) regarding the authors hypothesis of a ‘central pattern generator’ also needs to be clarified in light of this as well.

Authors: We addressed these comments and concerns several times before.

Another point is that the Reviewer states the spontaneous activity of the sinoatrial node cells. Indeed, it has been shown that SAN cells display spontaneous activity; however, to our knowledge, these findings are in preparations where the IcNS is also present. As such, this statement is weak, as spontaneous neuronal activity could drive the activity of the SAN cells. Furthermore, as the Reviewer states, the SAN cells are modulated by neurons, which in turn means that SAN cells are receiving input from neurons.

METHODS

- Lines 389-398

A wide array of zebrafish ages are used, which makes interpretation of the results difficult, and various findings possibly unrelated. Experiments using wild-type zebrafish used 8-12 week old animals (which are termed ‘Adult’, even though zebrafish have not yet reached breeding age/sexual maturity until 12 weeks, so is misleading), experiments with Tg(elavl3:eGFP) animals used 5-month and 11-month old animals (which are very different ages, yet it is unclear in which cases each were used, or if they were grouped together, which would be inappropriate), and experiments with Tg(nbt:DsRed) animals used 22-month old animals (which are in fact more ‘aged’ zebrafish). It is unclear why such a broad range of ages was used and which results relate to which age.

Authors: We apologize for any confusion caused by our previous presentation of this information. In all our studies, including anatomy, electrophysiology, and functional (*ex-vivo*) experiments, we exclusively utilized young adult zebrafish, as previously described (aged between 8-12 weeks). As we stated in the text before, older fish were specifically reserved for sequencing experiments due to technical challenges encountered when obtaining high-quality and viable neurons after dissociating the whole heart.

Determining the precise age of zebrafish (stage) using time, particularly beyond the larval stage, is inaccurate as zebrafish exhibit continuous allometric growth rather than time-dependent aging (Dutta, 1994). This means that the passage of time following fertilization or hatching is not an indicator of the animal's age; instead, size and body mass are accurate measures. Thus, a larger zebrafish is considered "older" than a smaller one, regardless of whether they hatched on the same day.

In our experiments, we employed animals with a total length (TL) exceeding 1.8-2.0 cm, a size determining the adult stage according to established criteria (Nüsslein-Volhard and Dahm, 2002; Westphal and O'Malley, 2013; McMenamin and Parichy, 2013). Zebrafish typically reach this stage within a span of 80-90 days (10-12 weeks), although this timeframe is approximate and influenced by various factors such as animal line, breeding, and maintenance conditions, as well as feeding regimens (McMenamin and Parichy, 2013). Moreover, within a given population of animals born at the same time, the size of the animals follows a Gaussian distribution. Therefore, we used animals exceeding 1.8-2.0 cm at all "ages" when available. Consequently, our animals are considered sexually mature because they are capable of successful breeding and thus meet the criteria for adulthood.

In light of the Reviewer's comment, we have revised the manuscript text to provide more detailed and explicit information regarding the age and stage of the experimental animals, ensuring clarity and accuracy in our presentation.

References:

- Dutta H. Growth in fishes. *Gerontology*. 1994;40(2-4):97-112. PMID: 7926860.
 McMenamin SK, Parichy DM. Metamorphosis in teleosts. *Curr Top Dev Biol*. 2013;103:127-65. PMID: 23347518.
 Nüsslein-Volhard C, Dahm R. (2002). *Zebrafish: A Practical Approach*. Oxford, England: Oxford University Press.
 Westphal RE, O'Malley DM. Fusion of locomotor maneuvers, and improving sensory capabilities, give rise to the flexible homing strikes of juvenile zebrafish. *Front Neural Circuits*. 2013 Jun 7;7:108. PMID: 23761739.

- **Lines 446-447**

- The concentration of blebbistatin reported to be used was 60 mM, unless this is a typographical error, this represents a concentration that is more 1000x greater than that reported by any other group (which all use concentration in the uM range). If a concentration that that high was truly used, blebbistatin could have secondary effects and the authors would need to report on this. Further to this, the authors report using GFP to identify intracardiac neurons for intracellular recordings, which would assumedly use a blue excitation light, which is known to inactivate blebbistatin. Could the authors comment on this?

Authors: Indeed, the concentration of blebbistatin we used was 60 μ M. We sincerely apologize for this mistake and thank the Reviewer for pointing it out. It is now corrected in the revised version of our manuscript.

Regarding the second point raised by the Reviewer, indeed, activation of the fluorescence, even at a lower intensity, reliably deactivates the effect of blebbistatin, and the heart starts beating. Thus, in our experimental procedure, we identified the neurons (GFP+) and marked their precise location before applying the blebbistatin and performing the recordings. To further verify that the recorded neuron was indeed a GFP+ neuron, we tested after the end of the experiment, and we imaged the neuron in both channels (DIC and GFP). We also include this information in the methods section of the revised manuscript.

- Electrodes for intracellular recordings had a resistance of 7-9 Mega Ohms, which is very low for such recordings, the authors need to provide an explanation/rationale for this.

Authors: We do not comprehend why our electrode resistance is too low for “such recordings”.

Electrode resistance is contingent upon factors such as the tip's diameter and the electrode's overall shape. This resistance, in conjunction with the resistance at the electrode-cell junction, collectively determines the access resistance. Elevated access resistance and subsequent current flow invariably translate into amplified **voltage error**, a phenomenon governed by Ohm's law.

It is imperative to recognize that high electrode resistance exacerbates access resistance, necessitating larger currents—typically at nanoampere (nA) levels—to elicit neuronal responses. This elevated current flow introduces inconsistencies in the measured membrane potential compared to the actual potential difference across the membrane. Furthermore, it amplifies thermal noise attributable to the motion of charged particles.

The ramifications of this inconsistency are profound, manifesting as distortions in recorded traces and action potentials. Such inaccuracies compromise the fidelity of electrophysiological measurements, undermining their reliability and interpretability. Hence, minimizing electrode resistance is paramount in ensuring the accuracy of electrophysiological readings.

The typical range of electrode resistance for patch-clamp recordings of neurons ranges between **5-15 M Ω** in the vast majority of studies/publications in many vertebrate species, such as lamprey (Kardamakidis et al., 2015), zebrafish (McLean et al., 2008; Gabriel et al., 2011; Ampatzis et al., 2013; Chang et al., 2020; Harmon et al., 2020), *Xenopus* (Zhang et al., 2012; Picton et al., 2018), and rodents (Dougherty et al., 2010; Calvigioni et al., 2023). This is also the case of non-CNS recorded neurons as the ones of the enteric nervous system (Hao et al., 2012; Hawkins et al., 2013) or the superior cervical ganglia (Amendola et al., 2015).

In a previous elegant report on heart neuron recordings in mammals, the authors used high-resistance electrodes (Harper and Adams, 2021); however, they did not perform patch-clamp recordings. They performed **sharp electrode intracellular recordings**, which required different types of electrodes (of higher resistance).

References:

- Amendola J, Boumedine N, Sangiardi M, El Far O. Optimization of neuronal cultures from rat superior cervical ganglia for dual patch recording. *Sci Rep.* 2015 Sep 24;5:14455. PMID: 26399440.
- Ampatzis K, Song J, Ausborn J, El Manira A. Pattern of innervation and recruitment of different classes of motoneurons in adult zebrafish. *J Neurosci.* 2013 Jun 26;33(26):10875-86. PMID: 23804107.
- Calvigioni D, Fuzik J, Le Merre P, Slashcheva M, Jung F, Ortiz C, Lentini A, Csillag V, Graziano M, Nikolakopoulou I, Weglage M, Lazaridis I, Kim H, Lenzi I, Park H, Reinius B, Carlén M, Meletis K. Esr1+hypothalamic-habenula neurons shape aversive states. *Nat Neurosci.* 2023 Jul;26(7):1245-1255. Epub 2023 Jun 22. PMID: 37349481.

- Chang W, Pedroni A, Hohendorf V, Giacomello S, Hibi M, Köster RW, Ampatzis K. Functionally distinct Purkinje cell types show temporal precision in encoding locomotion. *Proc Natl Acad Sci USA*. 2020 Jul 21;117(29):17330-17337. Epub 2020 Jul 6. PMID: 32632015.
- Dougherty KJ, Kiehn O. Firing and cellular properties of V2a interneurons in the rodent spinal cord. *J Neurosci*. 2010 Jan 6;30(1):24-37. PMID: 20053884.
- Gabriel JP, Ausborn J, Ampatzis K, Mahmood R, Eklöf-Ljunggren E, El Manira A. Principles governing motoneurons' recruitment during zebrafish swimming. *Nat Neurosci*. 2011 Jan;14(1):93-9. Epub 2010 Nov 28. PMID: 21113162.
- Hao MM, Lomax AE, McKeown SJ, Reid CA, Young HM, Bornstein JC. Early development of electrical excitability in the mouse enteric nervous system. *J Neurosci*. 2012 Aug 8;32(32):10949-60. PMID: 22875929.
- Harmon TC, McLean DL, Raman IM. Integration of Swimming-Related Synaptic Excitation and Inhibition by olig2⁺ Eurydendroid Neurons in Larval Zebrafish Cerebellum. *J Neurosci*. 2020 Apr 8;40(15):3063-3074. Epub 2020 Mar 5. PMID: 32139583.
- Harper AA, Adams DJ. Electrical properties and synaptic transmission in mouse intracardiac ganglion neurons in situ. *Physiol Rep*. 2021 Sep;9(18):e15056. PMID: 34582125.
- Hawkins EG, Dewey WL, Anitha M, Srinivasan S, Grider JR, Akbarali HI. Electrophysiological characteristics of enteric neurons isolated from the immortal mouse. *Dig Dis Sci*. 2013 Jun;58(6):1516-27. Epub 2013 Jan 31. PMID: 23371009.
- Kardamakis AA, Saitoh K, Grillner S. Tectal microcircuit generating visual selection commands on gaze-controlling neurons. *Proc Natl Acad Sci USA*. 2015 Apr 14;112(15):E1956-65. Epub 2015 Mar 30. PMID: 25825743.
- McLean DL, Masino MA, Koh IY, Lindquist WB, Fetcho JR. Continuous shifts in the active set of spinal interneurons during changes in locomotor speed. *Nat Neurosci*. 2008 Dec;11(12):1419-29. Epub 2008 Nov 9. PMID: 18997790.
- Picton LD, Sillar KT, Zhang HY. Control of Xenopus Tadpole Locomotion via Selective Expression of Ih in Excitatory Interneurons. *Curr Biol*. 2018 Dec 17;28(24):3911-3923.e2. Epub 2018 Nov 29. PMID: 30503615.
- Zhang HY, Sillar KT. Short-term memory of motor network performance via activity-dependent potentiation of Na⁺/K⁺ pump function. *Curr Biol*. 2012 Mar 20;22(6):526-31. Epub 2012 Mar 8. PMID: 22405867.

- Experiments were conducted at room temperature (~23°C), yet beating rates of the explanted hearts was ~3Hz – this is much higher than the heart rate reported by others at this sub-physiological temperature (<https://pubmed.ncbi.nlm.nih.gov/24671241/>; <https://pubmed.ncbi.nlm.nih.gov/30017908/>; <https://pubmed.ncbi.nlm.nih.gov/26730947/>) and some explanation needs to be provided by the authors.

Authors: Indeed, temperature has a notable impact on heart rate. As demonstrated in **Figure 5c**, the recorded heartbeat rates in our experiments fell within the expected range (from 114 to 214.8 bpm, average 173.1 bpm; see below **Figure 4 - for Reviewers**), as previously reported (Stoyek et al., 2022). However, the Reviewer anticipated a lower heart rate in our experiments conducted at room temperature (23-25°C). It is worth noting that a previously published study (Lin et al., 2014) reported a heart rate 5-10% lower at 24°C. Our data cover also this small reduction.

Several factors may contribute to the observed discrepancies:

Sample size: Variability among individual animals is inherent in any study. Furthermore, both male and female zebrafish were included in our dataset, despite potential differences in heart functionality between sexes (Hein et al., 2019). The composition of sexes within the dataset could influence sampling outcomes.

Anesthesia method: Unlike previous studies that employed MS-222 anesthesia, we utilized an ice-cold method for all functional experiments. This approach facilitates faster and more reliable recovery of heart rate post-dissection and isolation. Importantly, MS-222 affects both neurons and cardiomyocytes as it is a voltage-gated sodium channel blocker. While the fish recover from MS-222 anesthesia as an animal, the washout of the drug could be slower at a cellular level (Hedrick and Winmill, 2003; Ramlochansingh et al., 2014; Hsu et al., 2023), and one has to determine its potential long-term effect on the zebrafish heart's sodium channels (Treves-Brown, 2000; Hsu et al., 2023).

Variations in experimental conditions: Differences in heart rate between studies may also stem from varied breeding practices, housing conditions, and stress levels induced by handling procedures. Additionally, discrepancies may arise from the use of different zebrafish lines. In our functional studies, we employed elavl3 zebrafish, while other studies utilized wild-type AB zebrafish (Stoyek et al., 2022) or did not specify the exact zebrafish line (Lin et al., 2014). These genetic variations could contribute to subtle differences in heart rate.

Considering these factors, direct comparisons between studies are challenging. Therefore, we focused on analyzing our data using consistent and comparable methods.

Figure 4 - for Reviewers. An alternative presentation of the data in Figure 5c using the *ex-vivo* adult zebrafish preparation.

References:

- Hedrick MS, Winmill RE. Excitatory and inhibitory effects of tricaine (MS-222) on fictive breathing in isolated bullfrog brain stem. *Am J Physiol Regul Integr Comp Physiol.* 2003 Feb;284(2):R405-12. Epub 2002 Oct 31. PMID: 12414435.
- Hein S, Hassel D, Kararigas G. The Zebrafish (*Danio rerio*) Is a Relevant Model for Studying Sex-Specific Effects of 17 β -Estradiol in the Adult Heart. *Int J Mol Sci.* 2019 Dec 13;20(24):6287. PMID: 31847081.
- Hsu JC, Rairat T, Lu YP, Chou CC. The Use of Tricaine Methanesulfonate (MS-222) in Asian Seabass (*Lates calcarifer*) at Different Temperatures: Study of Optimal Doses, Minimum Effective Concentration, Blood Biochemistry, Immersion Pharmacokinetics, and Tissue Distributions. *Vet Sci.* 2023 Aug 24;10(9):539. PMID: 3775606.
- Lin E, Ribeiro A, Ding W, Hove-Madsen L, Sarunic MV, Beg MF, Tibbits GF. Optical mapping of the electrical activity of isolated adult zebrafish hearts: acute effects of temperature. *Am J Physiol Regul Integr Comp Physiol.* 2014 Jun 1;306(11):R823-36. Epub 2014 Mar 26. PMID: 24671241.
- Ramlochansingh C, Branoner F, Chagnaud BP, Straka H. Efficacy of tricaine methanesulfonate (MS-222) as an anesthetic agent for blocking sensory-motor responses in *Xenopus laevis* tadpoles. *PLoS One.* 2014 Jul 1;9(7):e101606. PMID: 24984086.
- Stoyek MR, MacDonald EA, Mantifel M, Baillie JS, Selig BM, Croll RP, Smith FM, Quinn TA. Drivers of Sinoatrial Node Automaticity in Zebrafish: Comparison With Mechanisms of Mammalian Pacemaker Function. *Front Physiol.* 2022 Feb 28;13:818122. PMID: 35295582.
- Treves-Brown K.M. *Applied Fish Pharmacology.* Kluwer Academic Publishers; Dordrecht, The Netherlands: 2000. pp. 209–210.

Point-by-Point Response to Reviewers comments:

We thank the reviewer for recognizing our effort to improve the previous version of the manuscript. We have further revised the manuscript accordingly, considering all comments and suggestions, as detailed below. The Reviewers' comments are indicated in bold and italics, while our responses are in regular style in this document. Moreover, the revised manuscript file highlights all the amended text to allow the Reviewers to track our changes.

Reviewer #1 (Remarks to the Author):

The authors have addressed my major concerns. This story will be useful for the field and for hypothesis generation.

Authors: We are grateful to the Reviewer for their insightful comments, which have significantly enhanced our study. We believe that our research could offer valuable insights into the field and inspire further studies.

Reviewer #2 (Remarks to the Author):

Overall, while the manuscript presents some very interesting and valid findings, there unfortunately continue to be some issues that preclude me from accepting this manuscript as it is:

Abstract: I can agree with the author's response and revision, acknowledging that my initial take on this may have been stylistic which can be subjective.

Introduction: The included revision to the use of zebrafish is much better, especially considering the target for a broader audience.

Authors: We would like to express our sincere gratitude to the Reviewer for dedicating time and effort to provide valuable comments aimed at enhancing the quality of our study. Despite any potential scientific debate, we are genuinely thankful for the Reviewer's in-depth and meticulous evaluation of our work, as well as their recognition of the study's significance in the field.

Clarification on surface versus superficial, while seemingly trivial, is appreciated as it does have different implications comparing zebrafish to mammalian hearts (again considering a broader audience).

Authors: We thank the reviewer for pointing out this potential issue in our text.

Line 122: While the word pacemaker may not have been used explicitly in the cited study, and the existence of CPG is not at all debated, there have been reports of rhythmic 'phase-locked' intracardiac neurons previously.

Authors: In our research, we set out to delve into the potential impact of the intracardiac nervous system on rhythm generation. Previous studies hinted at the possibility that the intracardiac nervous system (IcNS) could function as a pacemaker or as a network that could affect heart rhythms. By expanding upon this existing understanding, we aimed to provide further evidence that solidifies and supports this perspective. In the revised version of our manuscript, we acknowledge further these previous efforts. We consider that our work, in combination with the previous body of literature supporting the notion.

Lines 194-195: While agreeing with the possibility of continual regeneration and neurogenesis, and in fact it is very likely that the zebrafish heart possess a cardiac 'stem-cell'-like population that could result in changing neuronal numbers, the argument is not one of stability. If the

“zebrafish’s dynamic neurogenic and proliferative capabilities underscore the plausibility of ... developmental stages within its nervous system (at adult stages)”, how do the authors then fit their work into the scope?

Authors: Our research findings indicate the presence of a population of cells resembling stem cells in the zebrafish heart that possess potentially neurogenic capabilities. We are currently in the process of developing methodologies to more effectively study neurogenesis in zebrafish hearts, particularly in the context of injury and heart regeneration. Despite our ongoing efforts, a precise understanding of how the intracardiac nervous system responds to pathophysiological conditions still remains elusive. We anticipate that in the near future, we will have the ability to decode the cellular (neuronal) responses following injury and during heart regeneration. Nevertheless, the framework established by our study here sheds light on the various types of cells and neurons that may be involved in different physiological and pathophysiological states. We intentionally refrained from delving into detailed discussions on this aspect, as presenting preliminary findings and indications could potentially lead to misinterpretations. Hence, we kindly seek the reviewer's permission to omit this part from our manuscript.

Related to this is the total number of the number of neurons found in the current study, Stoyek et al. 2015 found ~200 HuC/D+ (labels neuronal cell bodies) and AcT (labels axons only) and the current study reports only 77. The authors say that only 30-40% in their study were HuC/D+ and that with the inclusion of HuC/D- the numbers would match, however if we are comparing only HuC/D+ only this does still does not make immediate sense. If corrected for age/development with (Stoyek et al. 2018, Prog Biophys Mol Biol) the numbers are closer (~150 total HuC/D+) but there still exists a discrepancy.

Authors: In addition to the previously mentioned factors (as per the Reviewer's earlier comment), such as the animal line and the definition of the SAP area, it is important to consider the age or stage as a potential explanation for the observed disparities in neuron numbers across the studies. Therefore, in our revised discussion, we elaborate on the age or stage-related variations in the intracardiac nervous system and provide a new citation to the research conducted by Stoyek et al. in 2018. As such, we consider that both the previous study (Stoyek et al., 2015, J Comp Neurol. 523, 1683-1700) and our study offer valuable and consistent information within their respective analytical frameworks.

Lines 233-234: I appreciate both the novel GABA/Glut findings, and the efforts to put things in better context.

Authors: We thank the reviewer for acknowledging our efforts.

In terms of the number of neurons found in the current study, Stoyek et al. 2015 found ~200 HuC/D+ (labels neuronal cell bodies) and AcT (labels axons only) and the current study reports only 77. The authors say that only 30-40% in their study were HuC/D+ and that with the inclusion of HuC/D- the numbers would match, however if we are comparing only HuC/D+ only this does still does not make immediate sense. If corrected for age/development with (Stoyek et al. 2018, Prog Biophys Mol Biol) the numbers are closer (~150 total HuC/D+) but there still exists a discrepancy.

Authors: As we stated above, we include more information in the revised version of the manuscript.

Lines 279-280 & 289-300: I appreciate the revision for clarity on the potential for a neuronal pacemaker operation and the ex vivo preparation.

Authors: Thank you.

Lines 348-350: My initial point here was simply that neurotransmitter content suggests function

but does not demonstrate it (which I think the authors also agree) and that the discussion of these findings should be worded as such.

Authors: Indeed, the neurotransmitter phenotype is a robust predictor of functionality in the intracardiac neuronal network. However, further functional studies, such as paired electrophysiological recordings, are essential to elucidate the precise role of each neurotransmitter within this complex network. Understanding the specific contributions of each neurotransmitter will provide valuable insights into the overall functionality of the intracardiac neuronal network. Accordingly, we revised our discussion to include this information.

Lines 370-373: The authors state that “SAN cells display spontaneous activity; however, to our knowledge, these findings are in preparations where the IcNS is also present. As such, this statement is weak..”. I would suggest that authors refer to Tessadori et al. 2012 (<https://doi.org/10.1371/journal.pone.0047644>), or other reports (especially those characterising the slo mo mutant) that show spontaneous depolarisation in isolated SANC from zebrafish, as well as others in mammals (<https://doi.org/10.1113/jphysiol.2003.040501>). While this does not exclude a potential neural contribution, it should be considered.

Authors: As suggested by the Reviewer, we revised the manuscript text accordingly to include the information above.

Lines 389-398: While I can agree that a precise developmental ‘staging’ with zebrafish can be problematic, the steps with TL are a great step towards that. However, even in zebrafish, with a gradual senescence (large body of literature on this), there is likely to be large differences at a cellular level between a 1.5–2.0cm zebrafish that is 8-12 weeks old versus one that is 22 months old. This again goes back to the comment on neuron numbers and the considerations for ‘developmental stages’ in terms of the nervous system.

Authors: We agree that potential differences can reflect the animals' age/stage. As we stated above, we include in the revised version of our manuscript a previous study (Stoyek et al., 2018) reporting the age-dependent differences in zebrafish.

Point-by-Point Response to Reviewers comments:

We thank the reviewer for recognizing our effort to improve the previous version of the manuscript. We have further revised the manuscript accordingly, as detailed below. The Reviewers' comments are indicated in bold and italics, while our responses are in regular style in this document. Moreover, the revised manuscript file highlights all the amended text to allow the Reviewers to track our changes.

Reviewer #2 (Remarks to the Author):

The authors have presented a work that has both confirmatory and novel components and represents a critical step in the study of the intracardiac nervous system, and does well to support the zebrafish as a relevant model to do so. I greatly appreciate the open dialogue with the authors on questions and clarifications that I have raised, as well as their open approach to answering these questions and framing their work in reference to the existing scaffolding of knowledge. At this point I feel that the authors have now addressed most all of my comments and concerns, however in the final reading I have one last minor clarification question that was somewhat raised by the authors their Revision 1 comments in terms of the neuron (ICN) identification.

Authors: We are sincerely grateful to the Reviewer for their incredibly insightful and valuable comments, which have significantly enhanced the quality of our study. We think that our research has great potential to provide valuable and meaningful insights into the field for further studies.

The authors stated that " Neurons expressing eGFP were identified and marked (position) using a microscope equipped with a CCD camera at 60x magnification. The heartbeat was stopped by adding the photosensitive myosin II inhibitor (\pm)-blebbistatin (60 μ M; Sigma-Aldrich, 203392) to the perfusion solution (in darkness). The contractile activity of the heart was abolished 20-30 minutes after, and whole cell patch-clamp intracellular recordings were performed, both in voltage- and current-clamp modes."

My two questions on this are:

1) It is known that ICN in the zebrafish embedded in the SAP tissues (myocytes and connective tissue), were any steps taken to expose the ICN, or a was a certain position (e.g., atrial lumen/SA valve leaflets exposed) found to offer better access? In either case, a brief statement of this would help future reproducibility by readers;

and the second, and more important question:

2) The authors state that ICN were marked before application of Blebbistatin. Was this a physical marker or a microscope software-based coordinates mark based on XYZ coordinates? Clarification on this is important for both reproducibility and to ensure that the authors are in fact recording an ICN and not accidentally from a SAN cell which are known to tightly surround the IcNS in zebrafish. The classification of the ICN subtypes is very important to the authors manuscript and while the SS/A/R ICN recordings show stable pre and post RMP, there is a bit of movement in the RMP of the (B)ursting type (though this could simply just be the representative images).

Authors: We thank the Reviewer for these technical comments on our electrophysiology approach. After placing the tissue in the recording chamber, we visually identified the GFP-expressing neurons on the tissue and centered them in the electrophysiology setup microscope's visually accessible area. Individual neurons and/or the area of the SAP neurons can be settled using precise coordinates as we use a motorized moving stage. When we attempt the recordings, the electrodes progress using

motorized manipulators toward the target neurons. We do not use any chemical but rather a mechanical way to penetrate the tissue and place the electrode close to the selected neuron.

Indeed, the lcN neurons are surrounded by other cells and tissue, as the Reviewer points out, which is in a way similar to what happens in our recordings from the central nervous system (Pedroni et al., 2024; Chang et al., 2021; Chang et al., 2020; Song et al., 2020; Bertuzzi et al., 2018; Bertuzzi and Ampatzis, 2018; Ampatzis et al., 2014; Song et al., 2016) where we have the meninges externally and within the tissue, we have many other cell types such as glia (astrocytes, oligodendrocytes, microglia), endothelial cells, and other neurons (not from the selected type). Thus, the surrounding cells and tissue can likely block the electrode tip during the penetration process, which can be evaluated visually and through significant changes in electrode resistance. In those cases, the recording attempt was aborted. However, the first usually unsuccessful attempts lead to superficial tissue damage that allows better electrode access later on. As such, we had numerous failed attempts in the process of obtaining our recordings. Also, we did not use any chemicals for the neuron recordings to aid the whole-cell patch-clamp mode, as it occurs during perforated patching. Here, the patching (approach, giga-seal, whole-cell) was performed by changing the pressure (positive, neutral, negative)

Moreover, we have a methodological routine for a post-hoc evaluation of the recorded neurons. In other words, we successfully recorded the desired neuron expressing a fluorescent protein. Similar to our spinal cord and brain recordings from selective neuronal populations expressing fluorescent proteins such as GFP, RFP, DsRed etc, following the recording, the withdrawal of the electrode will often damage the neuron, however in all cases (damaged or undamaged recorded cells), the exchange of the artificial intracellular solution with neuronal intracellular solution leads to significant reduction or even elimination of the fluorescent protein. As such, the targeted lcN neuron GFP is significantly affected. The post-hoc evaluation was performed on our lcN recordings with an apparent technical challenge, as upon activation of the light, the heart starts beating. Hence, the whole process had to be very fast. Yet, even if a heart movement exists, we can still see if the targeted neuron (GFP+) is still there or not.

Finally, regarding the RMP, we must point out that the neuron membrane potential is never completely stable in whole tissue recording (not as the neurons from isolation) as they receive constant input that can affect the membrane channels. As such, the RMP is determined from gap-free recordings of several seconds as the mean value (baseline activity) using the clampfit software. Moreover, the electrical properties of the neurons, such as input resistance and rheobase, can play a significant role in the “stability” of the membrane potential. As expected, neurons with high input resistance and low rheobase, such as the bursting neurons (see Figure 4d and Supplementary Figure 6), often have more “unstable” RMP as small currents can open their membrane channels, leading to membrane potential changes. At the same time, the less excitable neurons with low input resistance and high rheobase require more substantial input (of several pA) to change the membrane potential, and the RMP appears more “stable”.

Most of this requested information is technical tips and tricks that do not reflect the methods section of a research paper but rather a method/protocol-like paper. While we cannot accommodate all the technical information and procedure protocols here, we added some more information in the revised version of the methods that reflects the questions raised by the Reviewer.

Lastly, as part of the scientific community, we are always available to help and navigate other researchers who wish to perform similar experiments by providing useful technical information or demonstrating the procedures.

References:

- Pedroni A, Dai YE, Lafouasse L, Chang W, Srivastava I, Del Vecchio L, Ampatzis K. Neuroprotective gap-junction-mediated bystander transformations in the adult zebrafish spinal cord after injury. *Nat Commun*. 2024 May 21;15(1):4331. PMID: 38773121.
- Chang W, Pedroni A, Bertuzzi M, Kizil C, Simon A, Ampatzis K. Locomotion dependent neuron-glia interactions control neurogenesis and regeneration in the adult zebrafish spinal cord. *Nat Commun*. 2021 Aug 11;12(1):4857. PMID: 34381039.
- Chang W, Pedroni A, Hohendorf V, Giacomello S, Hibi M, Köster RW, Ampatzis K. Functionally distinct Purkinje cell types show temporal precision in encoding locomotion. *Proc Natl Acad Sci USA*. 2020 Jul 21;117(29):17330-17337. PMID: 32632015.
- Song J, Pallucchi I, Ausborn J, Ampatzis K, Bertuzzi M, Fontanel P, Picton LD, El Manira A. Multiple Rhythm-Generating Circuits Act in Tandem with Pacemaker Properties to Control the Start and Speed of Locomotion. *Neuron*. 2020 Mar 18;105(6):1048-1061.e4. PMID: 31982322.
- Bertuzzi M, Chang W, Ampatzis K. Adult spinal motoneurons change their neurotransmitter phenotype to control locomotion. *Proc Natl Acad Sci USA*. 2018 Oct 16;115(42):E9926-E9933. PMID: 30275331.
- Bertuzzi M, Ampatzis K. Spinal cholinergic interneurons differentially control motoneuron excitability and alter the locomotor network operational range. *Sci Rep*. 2018 Jan 31;8(1):1988. PMID: 29386582.
- Ampatzis K, Song J, Ausborn J, El Manira A. Separate microcircuit modules of distinct v2a interneurons and motoneurons control the speed of locomotion. *Neuron*. 2014 Aug 20;83(4):934-43. PMID: 25123308.
- Song J, Ampatzis K, Björnfors ER, El Manira A. Motor neurons control locomotor circuit function retrogradely via gap junctions. *Nature*. 2016 Jan 21;529(7586):399-402. PMID: 26760208.

A final, and very minor, note at this point also is that I believe (after being recently corrected) that EGFP is now the preferred abbreviation.

Authors: Thank you. We changed the abbreviation in our text and figures accordingly.